# Adverse impact of terrain steepness on thermally-driven initiation of orographic convection

Matthias Göbel[1,2], Stefano Serafin[3], and Mathias W. Rotach[1]

[1]Department of Atmospheric and Cryospheric Sciences, University of Innsbruck, Innsbruck, Austria
[2]Regional Office Salzburg and Upper Austria, GeoSphere Austria, Salzburg
[3]Department of Meteorology and Geophysics, University of Vienna, Vienna, Austria

**Correspondence:** Matthias Göbel (matthias.goebel@geosphere.at)

**Abstract.** Diurnal mountain winds precondition the environment for deep moist convection through horizontal and vertical transport of heat and moisture. They also play a key role in convection initiation, especially in strongly inhibited environments, by lifting air parcels above the level of free convection. Despite its relevance, the impact of these thermally-driven circulations on convection initiation has yet to be examined systematically. Using idealized large-eddy simulations ($\Delta x = 50$ m) with the WRF model, we study the effect of cross-valley circulations on convection initiation under synoptically undisturbed and convectively inhibited conditions, considering quasi-2D mountain ranges of different heights and widths. In particular, we contrast convection initiation over relatively steep mountains (20 % average slope) and less steep ones (10 %). One distinctive finding is that, under identical environmental conditions, relatively steep mountain ranges lead to a delayed onset and lower intensity of deep moist convection, although they cause stronger thermal updrafts at ridge tops. The temporal evolution of convective indices, such as convective inhibition and convective available potential energy, shows that destabilization over the steeper mountains is slower, presumably due to lower low-level moisture. Analysis of the ridgetop moisture budget reveals the competing effects of moisture advection by the mean thermally-driven circulation and turbulent moisture transport. In general, at mountaintops, the divergence of the turbulent moisture flux offsets the convergence of the advective moisture flux almost entirely. Due to the stronger ridgetop updraft, the mean advective moistening over the steeper mountains is higher; nevertheless, the total moistening is lower and the width of the updraft zone is narrower on average. Thus, buoyant updrafts over the steeper mountains are more strongly affected by the turbulent entrainment of environmental air, which depletes their moisture and cloud water content and makes them less effective at initiating deep convection. Saturated updrafts over less steep mountains, on the other hand, gain more moisture from the vapor flux at cloud base leading to significantly higher moisture accumulation. The lower entrainment rates in these simulations are revealed by the fact that equivalent potential temperature in the cloud decreases less strongly with height than over steeper terrain. The precipitation efficiency, a measure of how much of the condensed water eventually precipitates, is considerably larger over the less steep mountains, also due to lower total condensation compared with the steeper simulations. The relationship between mountain size and precipitation amount depends on the thermodynamic profile. It is nearly linear in cases with low initial convective inhibition, but more complex otherwise. The weaker convection over steeper mountains is a robust finding, valid over a range of background environmental stability and mountain sizes.

## 1 Introduction

Mountains are hotspots for the initiation of deep moist convection (DMC; Banta, 1990; Kirshbaum et al., 2018), which can result in thunderstorms with heavy precipitation in the form of rain and hail, lightning, and strong winds. The necessary (but not sufficient) ingredients for its onset (Doswell et al., 1996) are a conditionally unstable environment, enough moisture for clouds and precipitation to form, and a lifting or triggering mechanism.

In addition to synoptic-scale weather, mesoscale and boundary-layer processes in mountainous terrain can play an essential role in controlling these ingredients. Firstly, mountains act as a heat source that causes steep lapse rates in elevated mixed layers, which can be advected horizontally over nearby plains (e.g., Banacos and Ekster, 2010). Secondly, moisture often accumulates over mountains due to the convergence of diurnal mountain winds (e.g., Demko et al., 2009). These occur on different spatial scales (e.g. as slope, valley, or mountain-plain breezes) and are induced by differential heating or cooling of the atmosphere relative to adjacent regions (Zardi and Whiteman, 2013). In some cases, the simultaneous occurrence of an elevated mixed layer and low-level advection by the mountain-plain breeze creates optimal conditions for organized intense convection, such as supercell storms, at mountain foothills (Scheffknecht et al., 2017).

Besides transporting heat and moisture, diurnal mountain winds lead to uplift in convergence areas, providing a mechanism for the thermal forcing of orographic convection initiation (CI). Another orographic CI mechanism is forced uplifting of the airflow, which is a purely mechanical forcing that occurs irrespective of differential heating. Past research focused mostly on the latter process, as reviewed e.g. by Houze (2012) and Colle (2013).

Banta (1990) and more recently Kirshbaum et al. (2018) presented extensive overviews of mechanical and thermal forcing and their interaction. The impact of thermal circulations on CI can be especially large during periods of weak synoptic flow with large values of convective inhibition (CIN). Strongly inhibited environments require considerable lifting for CI to take place, so they delay or even completely prevent convection onset, even in the presence of large convective available potential energy (CAPE). This type of convection is referred to as non-equilibrium convection (Done et al., 2006), as opposed to equilibrium convection where CAPE is quickly consumed by DMC after its creation, for instance during the passage of a cold front. Non-equilibrium convection is associated with lower predictability, especially when convection is parameterized (Done et al., 2006, 2012; Zimmer et al., 2011; Keil et al., 2014).

The important and sometimes subtle role that diurnal mountain winds play in CI has been shown in several field campaigns and related modeling studies. Demko and Geerts (2010) simulated a convective day during the CuPIDO campaign (Damiani et al., 2008) in Arizona, confirming that DMC originated along a convergence line formed by the slope wind circulation, and moved downwind afterward. While the upslope flow provided the necessary moisture for DMC, it also cooled the air above the ridge. Thus, when upslope winds weakened after the convective episode and their advective cooling effect vanished, renewed convection was initiated after local destabilization (boundary-layer thermals reaching the level of free convection, LFC).

In the Convective and Orographically-induced Precipitation Study (COPS, Wulfmeyer et al., 2011) in southwestern Germany and eastern France, a large fraction of the observed convective events was triggered above mountainous terrain (the Black Forest) due to thermal forcing. A climatology of radar reflectivity in the same region, spanning eight summer seasons (Weckwerth

et al., 2011), revealed that twice as many convective events were initiated over the mountainous terrain than over the Rhine valley in the study period. On average, convection events also grew larger, more intense, and longer-lived over the mountains. The peak in convective activity was found to be around local noon, coincident with the period of most intense orographically-induced convergence. Manzato et al. (2022) came to similar conclusions in a 15-year climatology of cloud-to-ground lightning flash density over the Alpine area. They showed that lightning is less frequent on average over the main Alpine ridge than over the surrounding plains. However, they also demonstrated that CI takes place preferentially over elevated areas, more frequently in the afternoon, and that the higher flash density over plains is often due to the downstream propagation of mature storms that originated over the mountains.

In recent years, the field campaigns RELAMPAGO (Nesbitt et al., 2021) and CACTI (Varble et al., 2021) focused on understanding the mechanisms of CI and upscale development around the Sierras de Córdoba in Argentina. DMC frequently initiates here, aided by the convergence of upslope flows, and rapidly grows upscale to yield supercells and mesoscale convective systems (Mulholland et al., 2018), often resulting in large hail (Kumjian et al., 2020; Bechis et al., 2022). Marquis et al. (2021) found that the observable quantity that most clearly differentiates between cases of sustained and unsustained precipitation was the depth of the horizontal convergence zone. Nelson et al. (2022) designed idealized simulations to interpret cases where CI failed to occur in a supportive environment. They demonstrated that these events were affected by the relatively low relative humidity of the environmental air above the lifting condensation level (LCL), which favored the dilution of buoyant thermals by turbulent entrainment. The authors also found that, in the presence of converging thermally-driven upslope flows, much narrower boundary-layer thermals were sufficient to yield CI. Another modeling study related to RELAMPAGO showed that higher mountains resulted in stronger upslope flows and thus earlier CI (Mulholland et al., 2020).

The ability of diurnal mountain wind systems to lift air parcels above the LFC and thus trigger DMC depends on various governing parameters such as the surface energy balance, the environmental thermodynamic profile, background winds, and terrain geometry.

The surface energy balance controls the very existence of thermal circulations. The main drivers are solar forcing and soil moisture. The latter influences the partitioning of the available net radiation into latent and sensible heat fluxes. The vertical mass transport operated by orographic thermally-driven circulations depends on the balance between the energy input supplied by the surface sensible heat flux and the energy required to destabilize the atmospheric column (Leukauf et al., 2016). Stronger heat flux or weaker low-level stability imply increased turnover of the low-level atmosphere, and hence increased vertical transport of any trace constituent, such as moisture. However, if the moisture content is high enough for cloud formation, cloud cover alters the surface energy balance, weakening both surface fluxes and thermal circulations.

Besides stability, another property of the thermodynamic profile that received considerable attention in recent studies on orographic CI is relative humidity. Its vertical variability is thought to play a key role in controlling whether or not destabilization will occur (Kirshbaum, 2011; Nelson et al., 2022). High values of surface relative humidity lower the LCL and generally also the LFC, facilitating CI. Thermal circulations enhance the moisture content close to the surface by mass and thus moisture convergence. However, the surface-based vertical updrafts that compensate for low-level convergence lose moisture and buoyancy while ascending through a drier environment, because of turbulent mixing. In principle, shallow convection can lessen

the detrimental effects of dry entrainment by progressively moistening the environment during the course of the diurnal cycle, facilitating the transition towards DMC (Kirshbaum, 2011).

The presence and strength of background winds have multiple implications. Strong background winds intensify turbulence and entrainment, and they also vent away thermals before an organized updraft can form. They are thus in principle detrimental to CI at mountain tops (Kirshbaum, 2011). An environmental wind can also displace thermal updrafts from the mountaintop towards the lee side, creating an asymmetric pressure field that enhances moisture flux convergence at the leeward foothills (Panosetti et al., 2016). This process effectively shifts the preferential CI location leeward of the mountain tops, but it is sensitive to mountain height (it seemingly does not occur over shallow mountains). It is also sensitive to stratification, as documented by Hagen et al. (2011).

Besides wind speed, the wind direction relative to a mountain ridge matters as well. Crook and Tucker (2005) and Tucker and Crook (2005) found that vertical uplift, and thus the ability to initiate moist convection, is maximized over mountain ranges oriented along rather than across the incoming wind direction, because of the more favorable interaction between crosswise thermally-driven convergence and streamwise mechanically-forced displacement of the flow. Soderholm et al. (2013) pointed out that directional wind shear is important as well. Vertically uniform wind direction above a ridge implies that the warm and moist storm inflow will be eventually undercut by the gust front, leading to short-lived convection. If the wind blows along the ridge at low levels and across it higher up, the previously described negative feedback is disrupted, and the likelihood of sustained convection is increased.

Compared with the role of atmospheric properties, the impact of terrain morphology on CI has attracted much less attention. Imamovic et al. (2019) investigated the impact of terrain geometry on DMC under weak synoptic forcing and low CIN using idealized simulations with $\Delta x = 1$ km. They found a linear relationship between rain amounts and mountain volume in a surprisingly large portion of the governing parameter space—excluding situations with strong background winds or large mountains. This relationship only arises in a statistical sense, as the variability among equivalent model runs is enormous. Imamovic et al. (2019) explain the observed linear scaling as a consequence of larger mountains driving stronger thermal circulations, and they speculate that the relationship would break down in cases with relatively high CIN. Strongly inhibited conditions are by no means exceptional. For instance, according to long-term analyses of radiosoundings in the Alpine region (e.g., Manzato, 2003), persistent CAPE is not accessible due to high CIN for more than half of the time during the convective season.

This broad overview demonstrates that, although most of the controlling factors of DMC over mountains have been investigated rather in-depth, one particularly relevant knowledge gap still remains, namely the impact of terrain geometry on orographic CI in strongly inhibited conditions.

In this study, we use idealized large-eddy simulations (LES) to demonstrate the sensitivity of the slope wind system and the ridgetop moisture budget to the slope angle. We further explain the implications for the initiation and intensity of DMC and the additional impacts of mountain height and initial stratification.

## 2 Methods

We performed LES simulations with the Advanced Research (ARW) dynamical solver of the Weather Research and Forecasting (WRF) model version 4.5 (Skamarock and Klemp, 2008). The WRF model is an open-source community model suitable for multiple scales ranging from LES to global simulations. It integrates the fully-compressible, non-hydrostatic Euler equations in flux form with tendencies from subgrid-scale processes. The model uses a pressure-based vertical coordinate named $\eta$ with a smooth transition from terrain-following at lower levels to isobaric higher up (hybrid sigma-pressure vertical coordinate, Klemp, 2011). A horizontally and vertically staggered grid is used for spatial discretization, whereby the horizontal grid is of Arakawa-C type. By default, the advection is fifth-order in the horizontal and third-order in the vertical. For the integration in time, the WRF model uses a 3rd-order Runge-Kutta scheme and integrates acoustic modes on fractional steps (Wicker and Skamarock, 2002). Model configurations common to all our model runs are described in Sect. 2.1 and 2.2. The whole set of simulations, which comprises both dry and moist model runs and considers mountains of different widths and heights, is introduced in detail in Sect. 2.3.

### 2.1 Model configuration

Our discretization uses a horizontal grid spacing of $\Delta x = 50$ m and a time increment of $\Delta t = 1$ s. We use Weighted Essentially Non-Oscillatory (WENO, Liu et al., 1994) advection for momentum and scalar variables. We simulate moist airflow over a mountain ridge. Since we use periodic lateral boundary conditions, the ridge is infinitely long in the $y$-direction and repeated infinitely in the $x$-direction as a series of valleys and ridges. In the $x$-direction, the domain size equals the mountain width $w_\mathrm{m}$, which varies between simulations. In the $y$-direction, it is 40 km, which is enough to achieve robust along-valley averages of turbulence statistics and precipitation amounts, and a clear ordering of the simulations in terms of initiation time and intensity of deep moist convection.

In the vertical, there are 253 levels with $\Delta z$ between $\sim 20$ m near the ground and $\sim 100$ m from 12 km height to the model top at about 17 km height (details about the vertical grid are given in Appendix A). For the dry simulations (see Sect. 2.3), in which deep moist convection does not develop, we lowered the model top to 8 km with 102 vertical levels. Implicit Rayleigh damping (Klemp et al., 2008) with a damping coefficient of $0.2$ s$^{-1}$ is used above a height of 12 km (6 km for the dry simulations) to prevent vertically-propagating gravity waves from being reflected at the model top.

The effects of Coriolis force and curvature on the momentum budget are neglected, because of the small extent of the studied phenomenon and the associated high Rossby number. Subgrid-scale mixing is described with the well-established 1.5-order TKE scheme by Deardorff (1980). Turbulent fluxes in the surface layer are modeled with the revised MM5 similarity theory scheme (Jiménez et al., 2012). Microphysical processes are handled by the *Predicted Particle Properties* (P3, Morrison and Milbrandt, 2015) parameterization, a bulk scheme that complements the usual conservation equation for cloud ice mass with additional prognostic quantities, i.e., rimed mass, rimed volume, and number concentration.

To enable realistic interactions between the surface energy balance, surface winds, and precipitation, the setup of the land surface model and the radiative transfer parameterization required some attention. We used the NOAH land surface model

(Chen and Dudhia, 2001) and considered grassland land cover, with an albedo of 0.19 and a vegetation fraction of 80%. The soil texture type is loam. Radiation was modeled by an improved version of the Rapid Radiative Transfer Model for GCMs (RRTMG-K, Baek, 2017), setting the latitude to 47.7°N and the day of the year to 15 July for reasons explained in Sect. 2.3. As a simplification, neither orographic shading nor slope-dependent radiation was considered, leading to uniform surface forcing on slopes with different angles and orientations. For the moist simulations, this configuration led to domain-averaged diurnal peak values of net shortwave radiation, sensible heat flux, and latent heat flux, respectively, of $\sim$740 W m$^{-2}$, $\sim$130 W m$^{-2}$ and $\sim$430 W m$^{-2}$ (a Bowen ratio of about 0.3).

## 2.2 Initialization

All model integrations are started at 6 LT (local solar time), about one hour after sunrise, and run until 19 LT. The starting time is chosen to lie within the morning transition period, during which the reversal of slope wind direction occurs and the intensity of thermally-induced winds is low. To perturb the initial state and initiate convection, random potential temperature perturbations are introduced. The perturbations are uniformly distributed with a maximum amplitude of 0.1 K and applied equally to the lowest 12 model levels ($\sim$ 260 m) based on Kealy et al. (2019). As suggested by Kealy et al. (2019), we use vertically uniform initial perturbations, which are less susceptible to numerical dissipation during model spin-up and are therefore more effective at triggering convective circulations.

At model initialization, the temperature gradient at the interface between the ground and the atmosphere is set to zero by equating the skin temperature to the atmospheric temperature extrapolated to the ground. This ensures that the surface sensible heat flux at the beginning of the simulation is equal to zero at all grid points, thereby avoiding the development of spurious thermal circulations. The deep soil temperature varies with elevation in a manner compatible with climatological values in central Europe. The soil temperature is then linearly interpolated to the surface; the linear temperature profile in the soil ensures that also the ground heat flux is initially zero.

Following Schlemmer et al. (2011), the soil moisture saturation ratio increases quadratically from 60% at the surface to 75% at the lowest soil level at a depth of 1.5 m.

## 2.3 Simulation series

The initial thermodynamic profiles of all our simulations are shown in Fig. 1. They are based on an idealized version of a radiosounding from the intensive observational period (IOP) 8b of COPS. The radiosonde was launched on 15 July 2007 at 08:00 UTC (09:00 CET) from Burnhaupt Le Bas in the southern part of the Rhine Valley and characterizes the pre-convective environment. In the early afternoon of that day, a mesoscale convergence zone interacted with diurnal mountain wind circulations and enhanced low-level uplift, triggering deep moist convection over the Black Forest despite high CIN and only moderate CAPE values (Kalthoff et al., 2009). An idealized version of this sounding was first used by Kirshbaum (2011) and later by Panosetti et al. (2018). In this study, we introduce several modifications of the COPS IOP8b sounding, which ensure that the environment where ridgetop thermal plumes ascend has comparable properties across all simulations, despite differences in the terrain geometry.

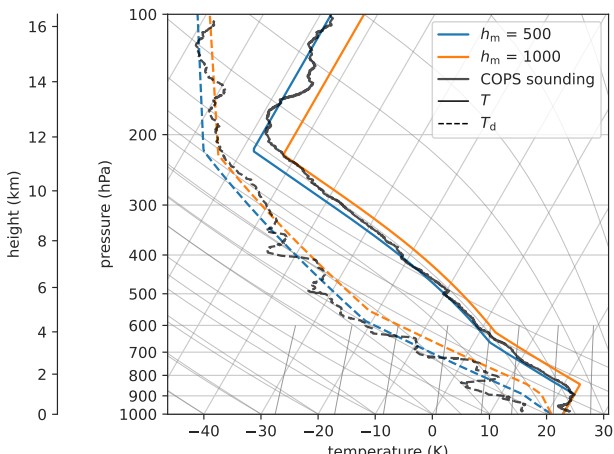

**Figure 1.** SkewT-diagrams of the initial temperature (solid lines) and dewpoint (dotted lines) profiles for the moist simulation series. The black lines are the temperature and dewpoint profiles of the observed COPS sounding that our cases are based on (see text in Sect. 2.3).

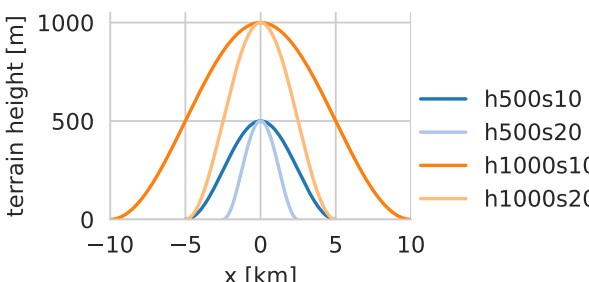

**Figure 2.** Orography of the performed simulations.

We prescribed the model orography with a cosine profile:

$$h(x) = \frac{h_{\mathrm{m}}}{2}\left[1 + \cos\left(\frac{2\pi x}{w_m}\right)\right] \tag{1}$$

and carried out simulations with different mountain heights $h_{\mathrm{m}}$ and widths $w_{\mathrm{m}}$. We used mountain heights of 500 m and 1000 m (low and high mountain) and average slope angles of 10 % and 20 % by changing $w_{\mathrm{m}}$ accordingly. The chosen values are roughly representative of the hilly terrain of the Black Forest region and are shown in Fig. 2 and Table 1. The terrain is homogeneous in the $y$-direction, which implies that no along-valley pressure gradient and no along-valley flow can develop in the simulations. This enables us to study the effects of cross-valley flows in isolation.

Table 1 gives an overview of all simulations and their characteristics. Initial profiles are designed so that CAPE, CIN, and the height of the LFC above the mountaintop are similar in all simulations. This ensures that the destabilization of the column

**Table 1.** Setup and characteristics of the moist simulations. The given convective indices refer to a 100 hPa mixed-layer parcel above the mountain ridge at model starting time. $h_s$ refers to the depth of the ground-based stable layer.

| terrain | $h_m$ (m) | avg. slope (%) | $w_m$ (km) | $h_s$ (m) | CIN (J kg$^{-1}$) | CAPE (J kg$^{-1}$) | LCL (m a.g.l.) | LFC (m a.g.l.) | LNB (km a.g.l.) | mean RH below LFC (%) |
|---|---|---|---|---|---|---|---|---|---|---|
| h500s10 | 500 | 10 | 10 | 1000 | 86 | 1075 | 1546 | 2376 | 11.4 | 54 |
| h500s20 | 500 | 20 | 5 | 1000 | 86 | 1075 | 1546 | 2376 | 11.4 | 54 |
| h1000s10 | 1000 | 10 | 20 | 1500 | 87 | 964 | 1612 | 2426 | 10.9 | 53 |
| h1000s20 | 1000 | 20 | 10 | 1500 | 87 | 964 | 1612 | 2426 | 10.9 | 53 |

**Table 2.** Initial temperature gradients. The initial surface temperature is 296 K.

| Height interval (m) | $\partial_z T$ (K km$^{-1}$) |
|---|---|
| $[0, h_m + 500]$ | 0 |
| $[h_m + 500, h_m + 3000]$ | $-8$ |
| $[h_m + 3000, 11500]$ | pseudo-adiabatic |
| $[11500, 17000]$ | 0 |

requires a similar energy input and that, hence, all simulations initially should have a similar chance of developing deep moist convection.

In all simulations a ground-based stable (isothermal) layer extends up to 500 m above the ridge, followed by a near-neutral layer up to 3000 m above the ridge, and a pseudo-adiabatic layer up to the tropopause at 11.5 km AMSL (see Table 2). The stratosphere has an isothermal stratification. The initial dewpoint profiles are also constructed relative to the mountain heights (see Table 3). In this way, the initial temperature and dewpoint profiles above the ridge as a function of height above ground are almost identical up to the tropopause for all simulations (not shown).

The whole set of simulations was repeated with greatly reduced moisture content to study the full diurnal cycle of the cross-valley circulation without interference from clouds and precipitation. For these dry simulations, the initial dewpoint is reduced by 14 K at all levels.

## 3 Results

### 3.1 Circulation intensity and convection initiation

For an overview of the cross-valley circulations that develop in the simulations, we first look at $y$-averaged cross-sections. Figure 3 shows water vapor mixing ratio, potential temperature, wind vectors, and cloud contours at 13 LT. The high-mountain

**Table 3.** Initial dewpoint gradients. The initial surface dewpoint is 294 K for the moist and 280 K for the dry simulations.

| Height interval (m) | $\partial_z T_d$ (K km$^{-1}$) | |
| | $h_m = 500$ m | $h_m = 1000$ m |
| --- | --- | --- |
| $[0, h_m]$ | $-7.3$ | $-3.8$ |
| $[h_m, h_m + 500]$ | $-6.8$ | $-7$ |
| $[h_m + 500, h_m + 4000]$ | $-10$ | $-10$ |
| $[h_m + 4000, 11500]$ | $-6.5$ | $-6.5$ |
| $[11500, 17000]$ | $-3$ | $-3$ |

simulations develop two stacked circulation cells, while the low-mountain simulations only have one large circulation cell. This feature is also found in simulations by Wagner et al. (2015), where stacked circulation cells appeared only for mountain heights of 1500 m and more. Likely, the critical mountain height for stacked circulations is dependent on valley width, stratification, and heat input. Intuitively, weaker stratification and stronger heat input lead to deeper mixing in a shorter time yielding a single
circulation cell. With a constant initial potential temperature gradient $\partial_z \theta_0$ of 3 or 4 K km$^{-1}$ (weaker stratification compared with the isothermal temperature profile) we also obtain single circulation cells only (not shown).

In Fig. 3 we see that the clouds develop faster in the s10 simulations. This is also visible in Fig. 4, which shows $y$-$z$ cross-sections along the ridge at 13 LT. In this figure, the inhomogeneity in $y$-direction due to turbulent eddies becomes evident. The cloud base, however, is rather homogeneous in the $y$-direction. The different thresholds for the cloud contour in Figs. 3 and 4
are due to the fact that $y$-averaging is applied in Fig. 3, but not in Fig. 4.

The s10 simulations feature an earlier onset of precipitation and much higher total precipitation sums than the steeper s20 runs (Fig. 5a). The h1000 simulations show larger precipitation totals than the h500 simulations. The h500s20 simulation has hardly any precipitation. The spatial distribution of precipitation is similar over the low and high s10 mountains although the high mountain is twice as wide as the low mountain: The precipitation is largest over the ridge and drops below 2 % of its
maximum value about 2 km away from it following a Gaussian curve (Fig. 5b). The data shown in Fig. 5 and all following figures are averaged in the $y$-direction. The shading indicates the variability (interquartile range) in the $y$-direction.

To understand the differences in convection initiation between the simulations, we first consider classical convective indices such as CAPE, CIN, LCL above ground level, and LFC above ground level for a 100 hPa mixed-layer parcel uplifted from the ridge top before precipitation onset (Fig. 6; lines in the figure terminate at the time precipitation reaches the ground). Diurnal
heating, erosion of the surface-based stable layer, and moisture convergence at the mountaintop imply that CIN, LCL, and LFC decrease, while CAPE increases during the course of the day. When the instability is gradually released in the form of cloud development, CAPE is reduced again (mostly visible for the h500 simulations).

By design, the initial values of the convective indices are about the same in all simulations. The time when CIN reaches values close to zero is generally well correlated with the precipitation onset. Only the h500s20 simulation does not follow this

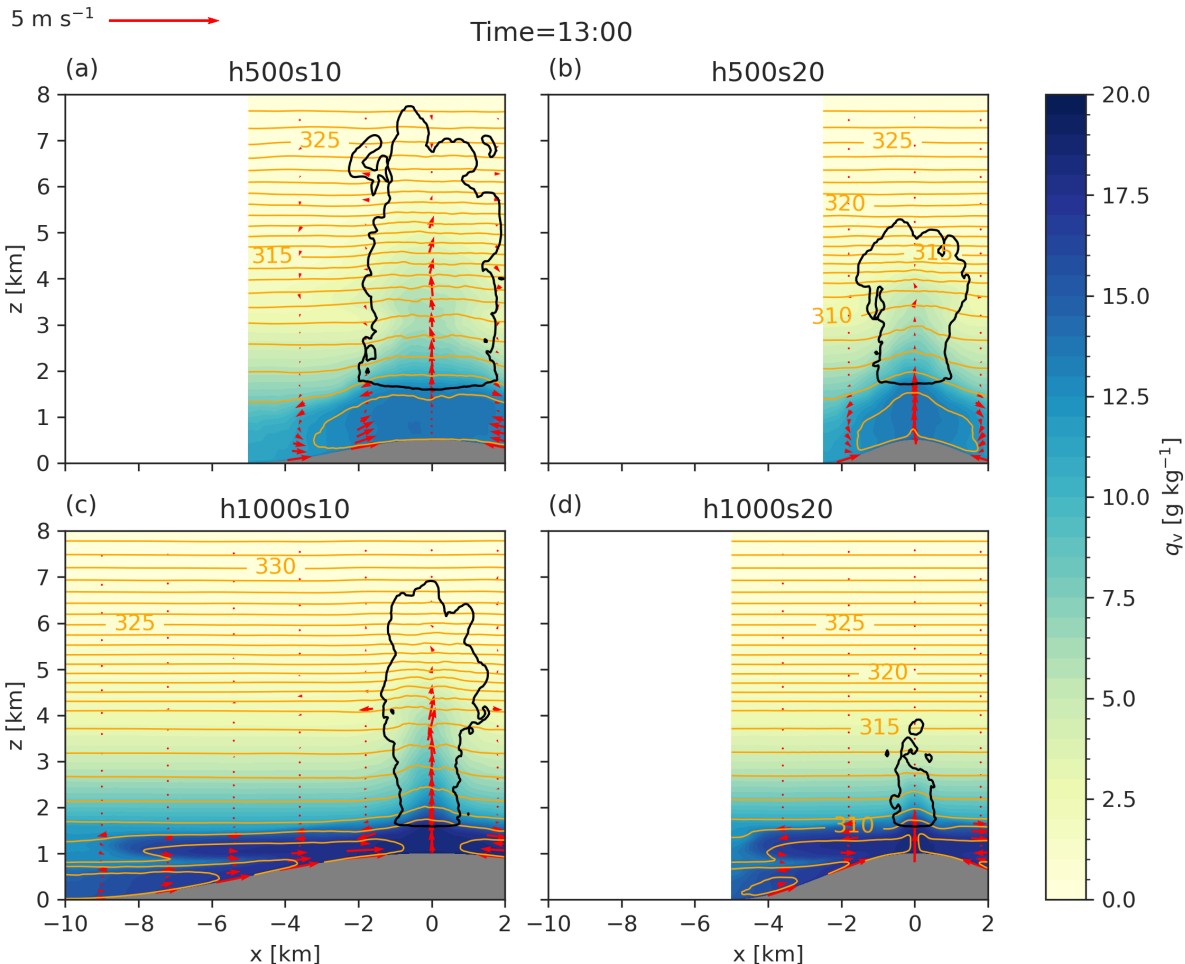

**Figure 3.** $y$-averaged cross-sections of the moist simulation series with water vapor mixing ratio (shaded), wind vectors, and potential temperature contours (spacing of 1 K) at 13 LT for (a) the h500s10, (b) h500s20, (c) h1000s10, and (d) h1000s20 simulations. The black contour line marks cloud water mixing ratio values above 1 mg kg$^{-1}$.

pattern, since it hardly produces any precipitation. This may be connected to the fact that this simulation, compared with all others, reaches the lowest maximum CAPE and highest minimum LCL and LFC throughout the day.

Precipitation onset follows CIN removal by a lag of one or two hours. The evolution of CAPE is also consistent with the precipitation onset: the s10 simulations feature a faster destabilization than the respective s20 simulations. The total precipitation can, however, not be deduced from CAPE alone: h1000s10 produces far more precipitation than h500s10, although its daily maximum CAPE value is only marginally larger.

When setting the dewpoint temperatures equal for the s10 and the respective s20 simulations, the evolution of the convective indices is nearly identical (not shown). Thus, the accumulation of moisture above the ridge seems to have an important influence

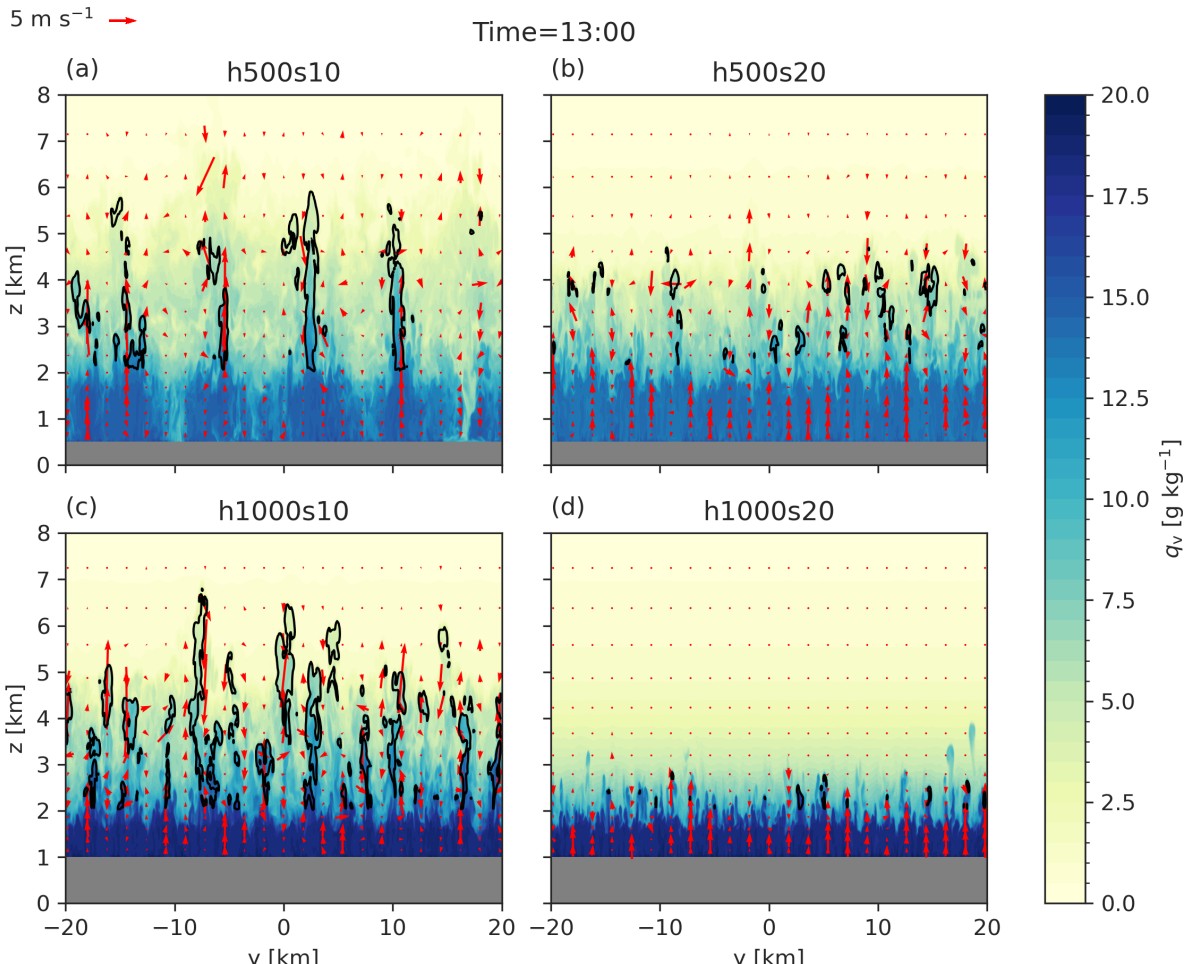

**Figure 4.** $y$-$z$ cross-sections along the ridge ($x = 0$) of the moist simulation series with water vapor mixing ratio (shaded) and wind vectors at 13 LT for (a) the h500s10, (b) h500s20, (c) h1000s10, and (d) h1000s20 simulations. The black contour line marks cloud water mixing ratio values above 1 g kg$^{-1}$.

on the time scale of convective destabilization. Therefore, we now examine the evolution of the cross-valley circulation and its impact on moisture transport towards the mountain ridge.

We examine the intensity of the cross-valley circulation as quantified by the strength of the updraft over the ridge, $w_{\mathrm{max}}$. Figure 7 shows the maximum vertical velocity in the vertical column above the ridge (one grid point in $x$-direction) for the dry and moist simulations. $w_{\mathrm{max}}$ is first calculated for each grid point in $y$-direction and then averaged in $y$-direction. As can be seen in Fig. 4, the averaging also includes downdraft regions, especially after cloud development starts. In the dry simulations, the circulation is stronger in the s20 runs almost all the time. In the moist simulations, we can see the sudden strengthening

of the updraft due to the latent heat released by cloud formation, which happens generally after 11 LT and earliest in the

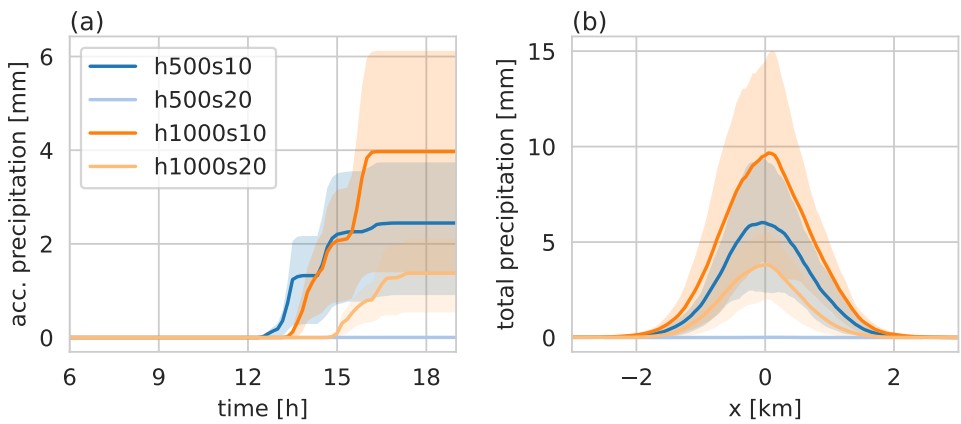

**Figure 5.** Accumulated precipitation in the four simulations (colored lines), (a) averaged in $y$ and $x \in [-2$ km, 2 km] as a function of time; (b) at simulation end, averaged in $y$ as a function of $x$. Shading indicates the variability (interquartile range) in the $y$-direction.

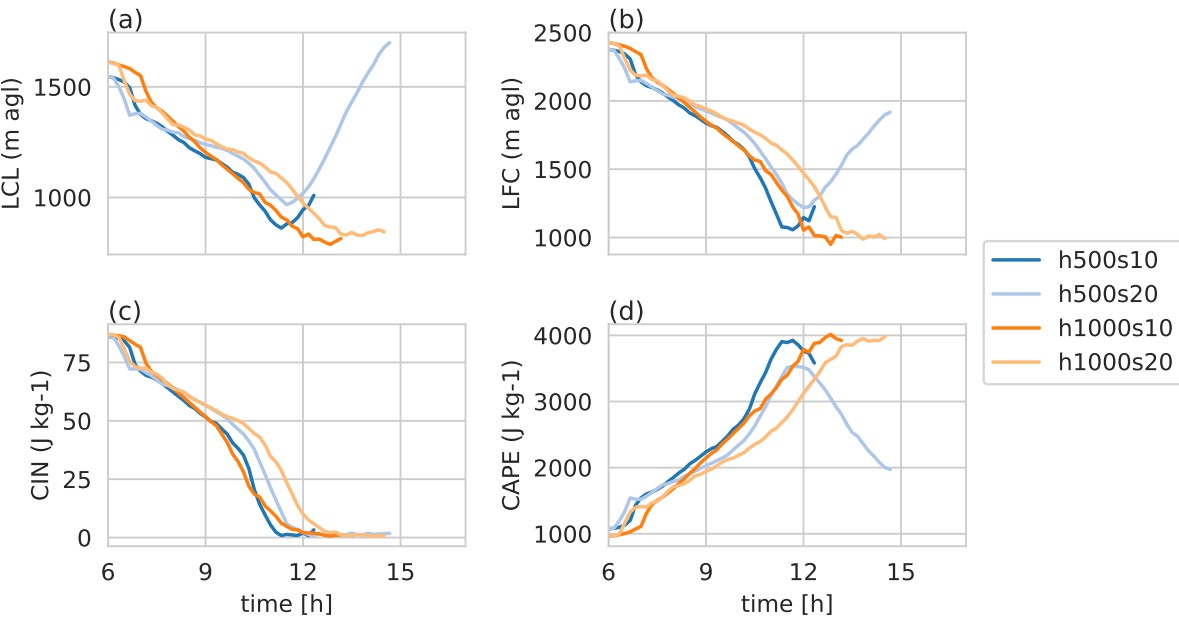

**Figure 6.** Evolution of (a) LCL (above ground level), (b) LFC (above ground level), (c) CIN, and (d) CAPE for 100 hPa mixed-layer parcels above the ridge before precipitation onset. Precipitation onset is defined as the time when the accumulated precipitation at the ridge ($x = 0$) reaches 0.01 mm.

h500s10 case. The h500s20 case that hardly developed any precipitation still shows a considerable increase in vertical velocity due to latent heating. The question remains, why the s20 simulations (over relatively steep terrain) develop a lower moisture

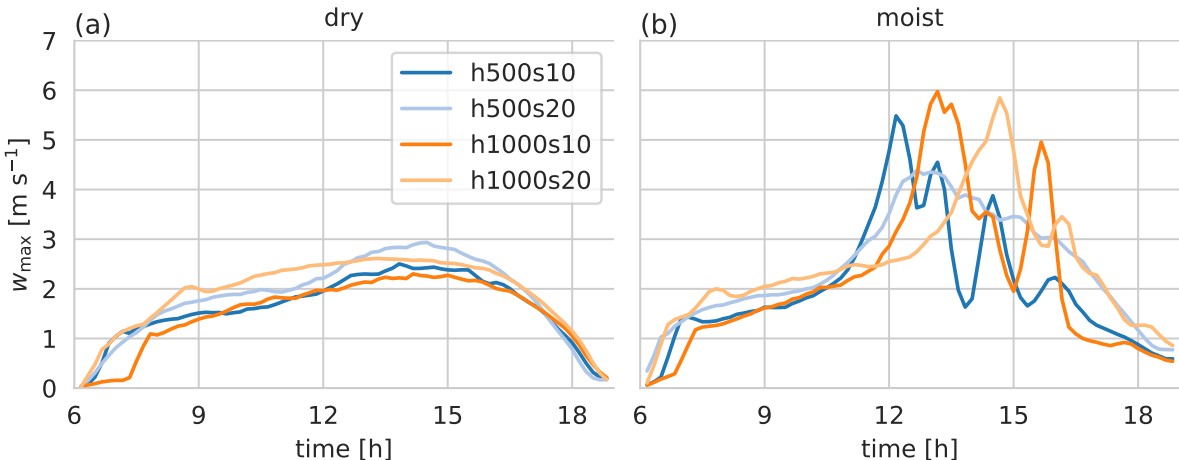

**Figure 7.** Time series of maximum vertical velocity in the vertical column above the ridge (one grid point in $x$-direction) averaged in $y$-direction for (a) the dry and (b) the moist runs (30-min moving average).

content above the ridge and thus later and weaker DMC compared with the s10 simulations, despite their thermal updrafts being persistently stronger in the dry simulations. To answer this question we investigate the moisture budget above the ridge.

### 3.2 Moisture budget

The budget calculations were performed with the WRFlux online tendency and flux averaging tool (Göbel et al., 2022), version 1.6.0 (Göbel, 2023b). All moisture budget components are averaged in time during model integration using 30-min block averages. In the postprocessing, the advection is decomposed into mean, resolved turbulent, and subgrid-scale turbulent components (see Schmidli, 2013; Göbel et al., 2022). We compute the cumulative change in $y$-averaged, vertically integrated water vapor content between the surface $z_0$ and the model top $z_{\text{top}}$:

$$\Delta Q(x,t) = \frac{1}{L_y} \int_{t_0}^{t} \int_{0}^{L_y} \int_{z_0}^{z_{\text{top}}} \frac{\partial \rho q_{\text{v}}}{\partial t} \, \mathrm{d}z \, \mathrm{d}y \, \mathrm{d}t' = Q_t - Q_0 \tag{2}$$

The budget consists of surface evaporation (not integrated vertically), mean advection, horizontal turbulent entrainment (resolved and subgrid-scale), and the microphysics tendency (evaporation + sublimation − condensation − deposition).

Figure 8 shows the moisture budget components for the dry simulations above the ridge ($x = 0$). The moistening due to surface evaporation is slightly higher for the h500 simulations, but no significant difference occurs between different slope angles (Fig. 8d). In contrast, the stronger circulation in the s20 runs leads to a stronger moistening due to the mean advective component compared with the s10 runs (Fig. 8a). The mean advective tendency is about one order of magnitude larger than surface evaporation. However, it is offset almost entirely by drying due to turbulent entrainment (Fig. 8b). The total moisture tendency is of the same order of magnitude as the surface evaporation and very similar for all simulations (Fig. 8f). Surface

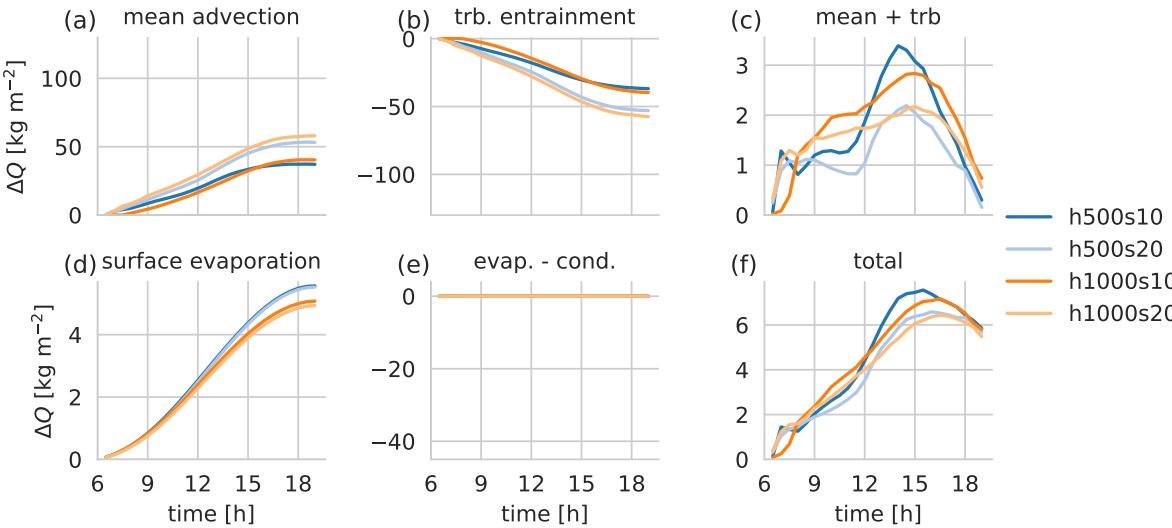

**Figure 8.** Time series of vertically and temporally integrated water vapor budget components above the ridge ($x = 0$) calculated with Eq. 2 for the dry runs. The panels show (a) mean advection, (b) turbulent entrainment, (c) sum of mean advection and turbulent entrainment, (d) surface evaporation, (e) net evaporation (evaporation - condensation), and (f) total moistening.

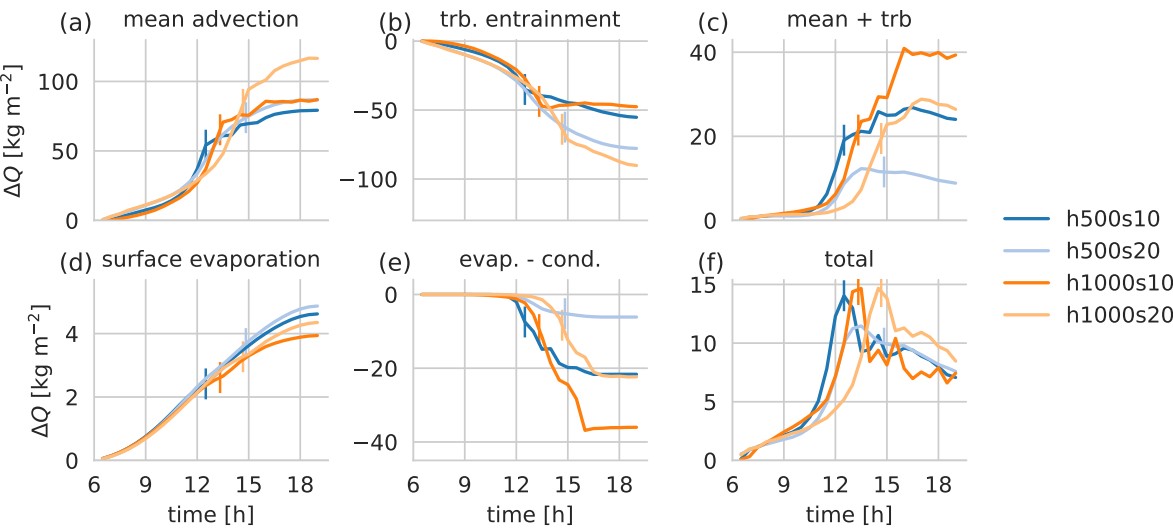

**Figure 9.** As Fig. 8 but for the moist runs. The vertical lines mark the precipitation onset, that is the time when the accumulated precipitation at the ridge ($x = 0$) reaches 0.01 mm.

evaporation amounts to between 12 and 30% of the total moistening in the early stages of the simulation, but the ratio increases to between 87 and 97 % towards the end of the day when the atmosphere is well mixed. Within the mixed layer, the total

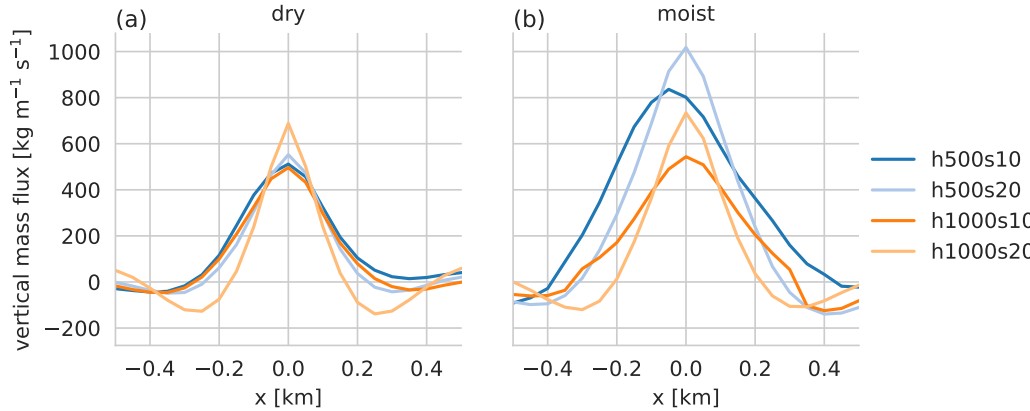

**Figure 10.** Vertical mass flux integrated vertically for (a) the dry and (b) the moist runs as a function of $x$ at 11 LT. For the moist runs the integration is only up to cloud base.

moistening is rather homogeneous in the vertical (not shown). The s10 cases show slightly more moistening than the s20 simulations which can at least partly explain why the convection starts earlier and is much more intense in the former.

The moisture budget for the moist runs is shown in Fig. 9. Due to cloud shading, the surface evaporation is slightly reduced compared with the dry runs. The net advective tendency (Fig. 9c) arising from the counteracting effects of mean advection and turbulent entrainment is much higher than in the dry runs. The resulting excess moisture is partly compensated by net condensation starting around noon (Fig. 9e). Overall, there is a strong gain in moisture content above the ridge (Fig. 9f) due to the strengthening of the circulation after 11 LT when clouds start to form (Fig. 9a), followed by a decrease in moisture due to continued condensation (Fig. 9e) and the disruption of the cross-valley circulation by precipitation (marked with vertical lines in Fig. 9). The residual of the budget (absolute difference between the sum of all forcing terms and the actual model tendency) never exceeds 2.7 % of the total moistening for any simulation and is thus far below any of the budget components.

So far, the strength of the circulation was only quantified in terms of the updraft velocity directly over the ridge. Figure 10 shows the $y$-averaged vertically integrated vertical mass flux at 11 LT as a function of $x$. In the moist simulation, the integration is only up to the average cloud base.

These diagrams show the horizontal extent of the updraft zone on top of the mountains. Properties of the updraft zone should not be misinterpreted as properties of individual updrafts. A wider ($y$-averaged) updraft zone does not necessarily mean that the individual updrafts are actually wider in that simulation, but rather that they occur more frequently or are stronger in the vicinity of the ridgetop.

The updraft zone is broader over the s10 mountains than over the respective s20 mountains. A possible explanation is connected with the variation of static stability with height. In all our simulations, static stability is higher at low levels, over the valley. Therefore, updraft development is favored at higher altitude, over the ridges. Intuitively, the s10 terrain provides a

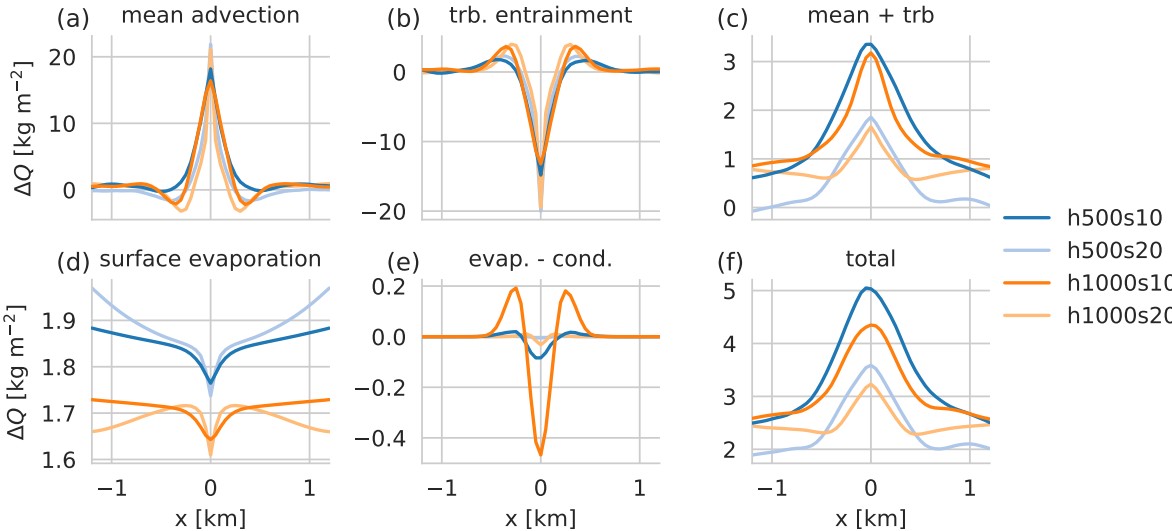

**Figure 11.** As Fig. 9 but as a function of $x$ at 11 LT.

larger area above a certain height than the s20 terrain, offering more room for thermals to develop, and ultimately leading to broader updraft zones.

As cloud formation and latent heat release already started at 11 LT (Figs. 9e and 13), the updraft zone is somewhat wider and
300 stronger in the moist simulations compared with the dry ones. With the wider updraft zones in the s10 runs, more moisture is transported upwards, eventually leading to higher total precipitation compared with the s20 runs. In addition, the wider updraft zones might make the individual updrafts less susceptible to moisture detrainment, because the air in between them will be moistened more effectively (by detrainment from previous updrafts) than the environmental air at greater distance from the ridge.

The narrower updraft zone over the s20 mountains can also be seen in Fig. 11, which shows again the vertically and temporally integrated water vapor budget components for the moist runs now as a function of $x$ at 11 LT. At mountaintop ($x = 0$), the mean advective moistening and the drying due to turbulent entrainment are stronger in the s20 than in the respective s10 simulations. However, already one gridpoint away from the ridge, these budget components are stronger in the s10 simulations. The total moisture tendency is considerably larger for the s10 mountains everywhere in the domain, owing to the more favor-
able balance between mean advection and turbulent entrainment. h500s10 has the largest total moistening at 11 LT which is connected with it having the earliest precipitation onset (see Fig. 5).

We therefore hypothesize that the earlier onset and higher amount of precipitation observed in the s10 runs are linked with the size of the updraft zones. If the whole updraft zone is considered, the rate of advective moisture transport over the ridge in the s10 runs is considerably higher. Broader updraft zones, which form preferentially over the less sharp orographic profile of
315 the s10 runs, are thus more likely to evolve into precipitating cumulonimbus clouds.

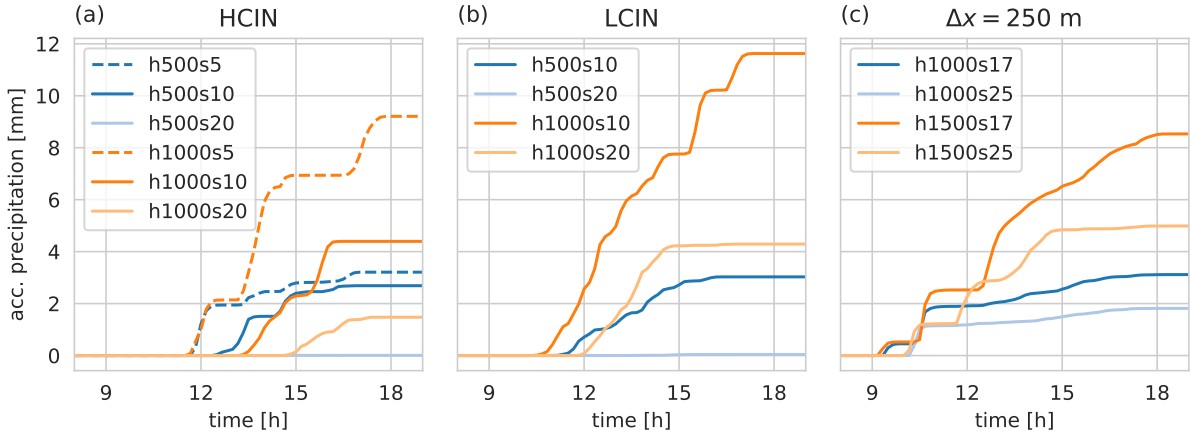

**Figure 12.** As Fig. 5a, but for (a) the already examined runs with high initial CIN (HCIN), including two additional simulations with 5 % average slope, (b) runs with low initial CIN (LCIN) and (c) runs with adjusted mountain height $h$ and slope $s$ and adjusted initial profiles so that the mountain protrudes the stable layer. Simulations in (c) were run with $\Delta x = 250$ m.

## 4  Discussion

In this section, we discuss the robustness of our findings with respect to some aspects of the initial conditions (Sect. 4.1) and investigate the precipitation process in more detail. For the latter purpose, we assess precipitation efficiency (Sect. 4.2) and determine how the precipitation scales with the mountain volume (Sect. 4.3). We also link these findings to previous literature.

### 4.1  Sensitivity to convective inhibition and a broader range of mountain geometries

To estimate the robustness of our results, we performed two more simulations with an even lower average slope angle of 5 % and with the same initial sounding as the h500 and h1000 simulations, respectively. We also ran some simulations with the same terrain configuration as before but a less stable stratification. We changed the temperature gradient in the first layer from isothermal to -3 K km$^{-1}$ and in the second layer to -7.1 K km$^{-1}$. For the h500 simulations, the surface temperature in the valley is 296 K, as before. For the h1000 simulations, it is raised to 297.5 K to obtain the same temperature at the mountaintop as in the h500 simulations. The dewpoint profiles remain the same as before. With this new stratification, initial CIN is reduced to about 65 J kg$^{-1}$, while initial CAPE is reduced by about $(160 \pm 10)$ J kg$^{-1}$. Therefore, we call these simulations the low CIN (LCIN) simulations and the original ones the high CIN (HCIN) simulations.

We also investigated cases where the model orography corresponds more closely to typical alpine mountain ranges. These are generally higher and steeper than the Black Forest hills, which inspired the simulations described so far. In these additional cases, the mountain ridge reaches 500 m above the stable layer using mountain heights of 1000 m and 1500 m combined with average slopes of about 17 and 25%. The model is run with $\Delta x = 250$ m. For brevity, we omit details about the model settings and the initial profiles. The namelist files and input soundings are available on Zenodo (Göbel, 2023a).

The additional simulations described in this section are not listed in Table 1 but the nomenclature for the terrain (consisting of mountain height X and slope Y: hXsY) is the same.

Figure 12a shows the accumulated precipitation for all the HCIN simulations. Precipitation starts earlier and reaches higher total values in the s5 runs compared with the respective s10 and s20 runs, which is consistent with our general hypothesis of earlier and stronger convection for less steep mountains. As expected, precipitation starts earlier in the four LCIN (between 11 and 12 LT, Fig. 12b) than in the respective four HCIN simulations (between 12 and 15 LT). The total amount of precipitation is increased, especially for the h1000 simulations. The h500s20 case again produces hardly any precipitation. Thus, the conclusion that the steeper mountains lead to later and weaker DMC is also valid for the more unstable simulations.

The circulation intensity before cloud development in the LCIN simulations is similar to the HCIN simulations, with a stronger but narrower updraft for the steep runs. The updrafts are stronger and wider compared with the HCIN runs (not shown).

Figure 12c shows the accumulated precipitation for the higher and steeper simulations with $\Delta x = 250$ m. Also in these simulations the conclusion still remains the same: steeper mountains lead to later and weaker convection.

Another common feature of all simulations is that, at constant slope angle, the higher mountain simulations produce higher precipitation totals. We will address this in Sect. 4.3.

## 4.2 Cloud water budget and precipitation efficiency

In Sect. 3 we demonstrated that orographic convection over mountains of lower steepness is connected with an earlier onset and a higher accumulation of rainfall. We interpreted this finding as a consequence of a wider updraft zone and a more favorable balance between moisture advection due to the thermally-driven circulation and moisture diffusion due to turbulence, which results in a stronger accumulation of water vapor at the mountaintop. This finding is in line with recent research emphasizing that dry air entrainment plays a key role in determining the fate of moist updrafts (Nelson et al., 2022; Marquis et al., 2021; Kirshbaum, 2011). Here we demonstrate that updraft dilution ultimately results in a reduction of the precipitation efficiency, i.e., the fraction of condensed water mass that actually precipitates to the ground (Demko and Geerts, 2010). For this purpose, we complement the preceding treatment of the water vapor budget (Sect. 3) with an analysis of the cloud water budget.

Following Demko and Geerts (2010) we specify a cloud control volume, wherein we perform a total water (vapor, liquid, and solid) budget analysis at hourly intervals. The definition of the cloud control volume is based on the average water vapor tendency by the microphysics scheme (evaporation + sublimation − condensation − deposition) within the last hour. The vertical boundaries are defined along $\eta$-levels (vertical model levels in WRF) instead of constant height levels to avoid numerical inaccuracies in the budget caused by interpolation. The control volume spans the whole simulation domain in the $y$-direction and its cross-section corresponds to the smallest rectangle (in the $x - \eta$ plane) containing all grid points with $y$-averaged microphysics tendency less than $-10^{-8}$ s$^{-1}$ (significant net condensation/deposition). Inactive cloud parts (no significant net condensation/deposition) that have been advected away from the cloud core region are thus excluded. Water vapor and hydrometeors can be transported laterally across the cloud volume boundaries by advection and turbulent exchange. The cloud

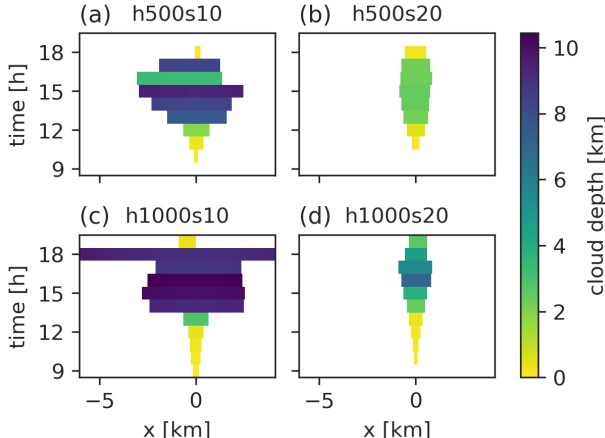

**Figure 13.** Depth and horizontal extent of the active cloud developing with time for (a) the h500s10, (b) h500s20, (c) h1000s10, and (d) h1000s20 simulations.

control volume varies in time, both horizontally and vertically. In contrast, Demko and Geerts (2010) fixed the control volume horizontally and only let the vertical boundaries change with time.

The cloud volume grows in time horizontally and vertically as shown in Fig. 13, most noticeably in the s10 simulations. The cloud volume is much narrower in the s20 runs, also in the early stages of cloud development. This is in line with the narrower updraft zone in these runs, as discussed earlier. Figure 14 illustrates the definitions of the active cloud volume and the different cloud water budget components. The cloud volume gains water by the vertical vapor flux at cloud base (VVF$_b$) and loses water due to the horizontal fluxes of vapor (HVF) and hydrometeors (HHF), the respective vertical fluxes at cloud top (VVF$_t$ and VHF$_t$), and precipitation that leaves the cloud volume (P). Part of this precipitation evaporates before reaching the

ground (E$_{PBL}$). The surface precipitation is thus approximately $|P| - E_{PBL}$. The total water tendency is denoted with $\Delta Q_t$. $\Delta Q_t$ also contains the change in total water content due to the change of the size of the cloud volume alone ($\Delta Q_t^{VC}$). This component is also treated as a source (or loss) term for $Q_t$.

We computed the different budget components from the output of our budget analysis tool WRFlux (HVF, VVF$_{b/t}$, E$_{PBL}$) and from standard WRF output (P, $\Delta Q_t$, $\Delta Q_t^{VC}$). In contrast to Demko and Geerts (2010), we accumulated the values over

380 the whole simulation time instead of using hourly aggregates since in our case the precipitation is considerably delayed with respect to the moisture accumulation. In the $y$-direction, all components are averaged, not integrated, since the domain length in the $y$-direction is arbitrary. Therefore, the budget components are given in kg m$^{-1}$.

E$_{PBL}$ is computed by integrating the $y$-averaged net evaporation rate E (in kg m$^{-3}$ s$^{-1}$) below cloud base:

$$E_{PBL} = \int_{t_0}^{t} \int_{x_1}^{x_2} \int_{z_0}^{z_1} E \, dz \, dx \, dt' \tag{3}$$

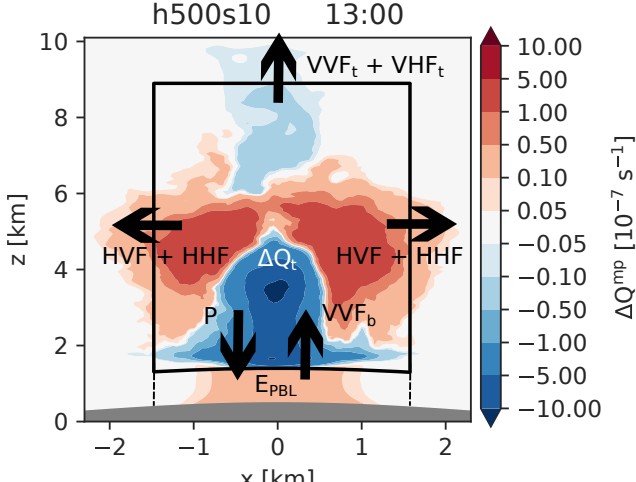

**Figure 14.** Concept of the cloud water budget. The shading indicates the average water vapor tendency by the microphysics scheme between 12 and 13 LT in the h500s10 simulation. The black box is the active cloud volume. The cloud water budget (vapor, liquid, and solid) consists of the horizontal vapor flux HVF, the vertical vapor fluxes at cloud base and top $VVF_b$ and $VVF_t$, the horizontal hydrometeor flux HHF, the vertical hydrometeor flux at cloud top $VHF_t$, the precipitation that leaves the cloud volume P, and the total change of cloud water $\Delta Q_t$. $E_{PBL}$ denotes the precipitation that evaporates below cloud base (between the dashed lines).

where $x_1$ and $x_2$ denote the left and right boundaries of the cloud volume, respectively, and $z_1 = z_1(x, \eta_1)$ is the height of the lower boundary (height of model level $\eta_1$ at $x$).

Since the fluxes of hydrometeors are not tracked in WRFlux, we deduced HHF and $VHF_t$ from two hydrometeor mass budgets $\partial_t Q_H$ outside the cloud volume. The only relevant terms in these budgets are HHF ($VHF_t$) and the net evaporation left and right of (above) the cloud volume. Hence,

$$
\text{HHF} = -\int_{t_0}^{t} \int_{-L_x/2}^{x_1} \int_{z_0}^{z_{\text{top}}} (\partial_t Q_H + E)\, dz\, dx\, dt'
$$

$$
-\int_{t_0}^{t} \int_{x_2}^{L_x/2} \int_{z_0}^{z_{\text{top}}} (\partial_t Q_H + E)\, dz\, dx\, dt' \tag{4}
$$

and

$$
\text{VHF}_t = -\int_{t_0}^{t} \int_{x1}^{x_2} \int_{z_2}^{z_{\text{top}}} (\partial_t Q_H + E)\, dz\, dx\, dt' \tag{5}
$$

where $z_2 = z_2(x, \eta_2)$ is the height of the upper boundary of the cloud volume and $L_x$ is the domain width.

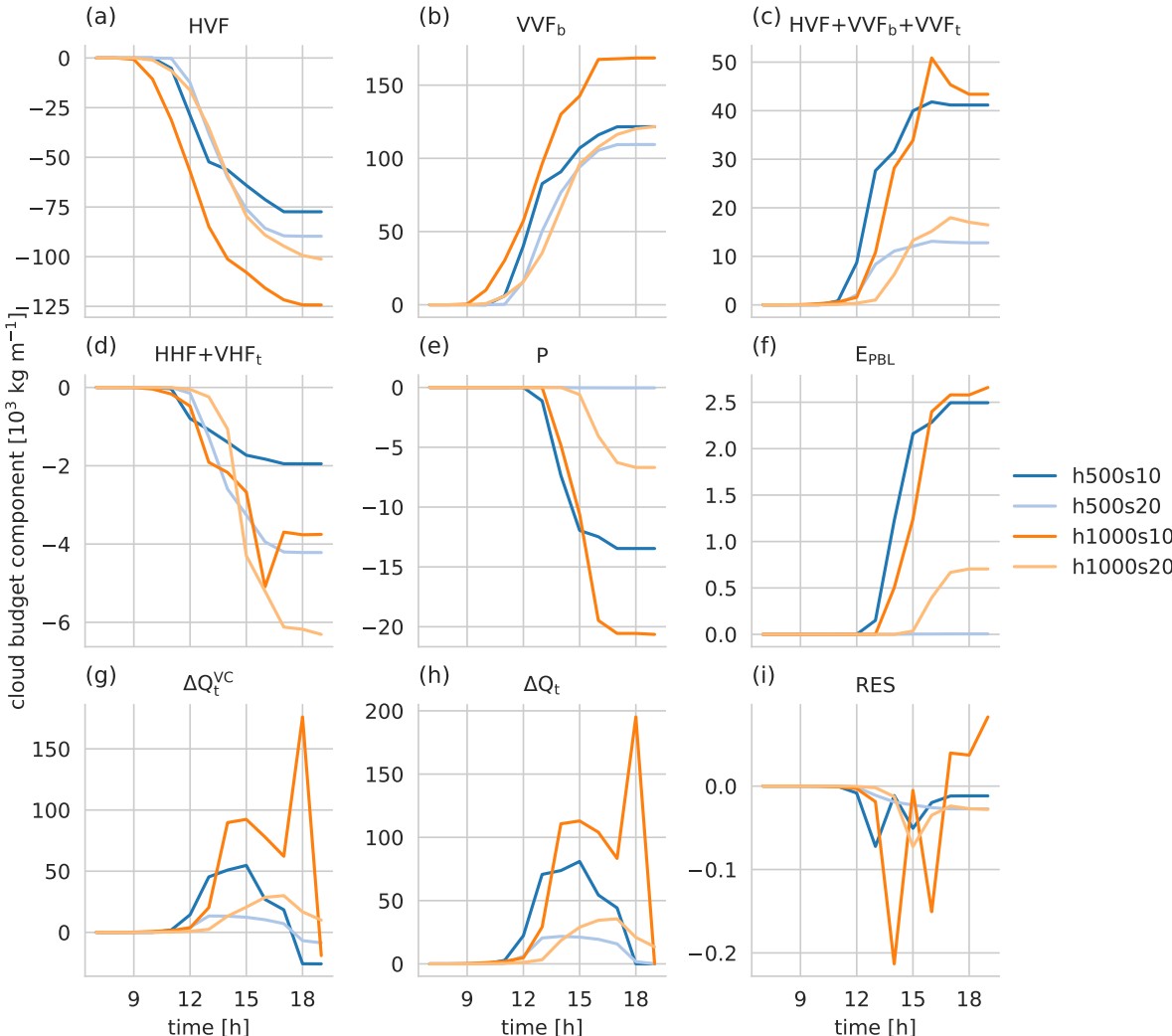

**Figure 15.** Timeseries of the accumulated cloud water budget components. Positive values indicate a gain of total water (vapor, liquid, and solid) in the cloud volume. The panels show (a) the horizontal vapor flux HVF, (b) the vertical vapor flux at cloud base $VVF_b$, (c) the sum of horizontal and vertical vapor fluxes $HVF+VVF_b+VVF_t$ ($VVF_t$: vertical vapor flux at cloud top), (d) the sum of horizontal and cloud-top vertical hydrometeor fluxes $HHF+VHF_t$, (e) the precipitation that leaves the cloud volume, P, (f) the precipitation that evaporates below cloud base $E_{PBL}$, (g) the change of cloud water due to volume change $\Delta Q_t^{VC}$, (h) the total change of cloud water $\Delta Q_t$, and (i) the residual RES.

The temporally accumulated cloud water budget components are shown in Fig. 15. All budget components are defined as being positive when they are a gain for total water in the cloud volume. RES denotes the residual of the budget, i.e.

$$RES = \Delta Q_t - (HVF + HHF + VVF_b + VVF_t$$
$$+ VHF_t + P + \Delta Q_t^{VC}). \tag{6}$$

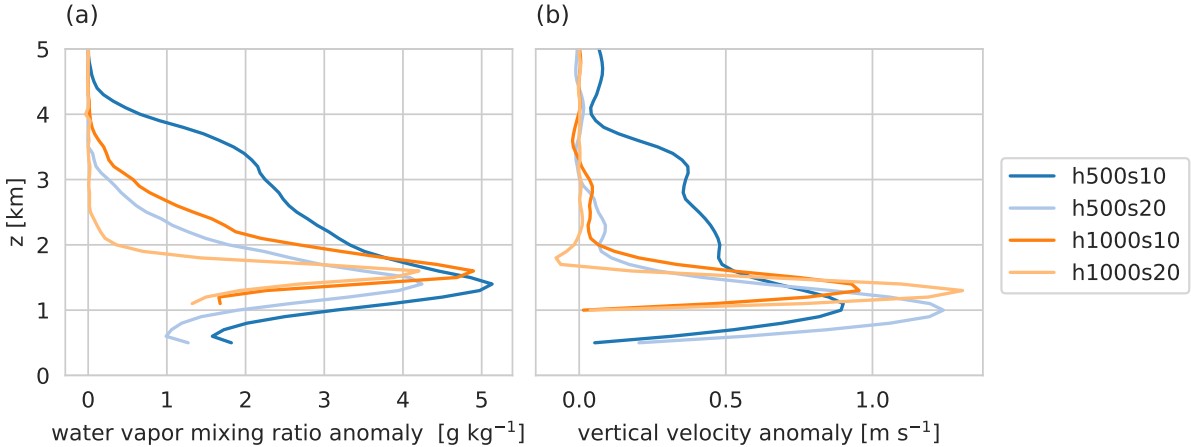

**Figure 16.** Profiles of water vapor mixing ratio anomaly (a) and vertical velocity anomaly (b) of the active cloud with respect to the surrounding environment in the moist runs at 12 LT.

RES is negligible compared with the other shown components (Fig. 15i).

The main source term $VVF_b$ is to a large extent compensated by HVF. $VVF_t$ is also negative but with a relatively small magnitude since vertical velocities are small at cloud top (not shown). The s10 mountains drive a stronger cloud base moisture flux than the respective s20 mountains. Therefore, and in the case of the h500s10 mountain also due to a decreased outflow of HVF later in the day, the net advective water vapor tendency ($HVF + VVF_b + VVF_t$) at the end of the day is between 2.6 and 3.2 times as large for the two s10 mountains compared with the two s20 mountains (Fig. 15c). About 10 to 20 % of the precipitation that leaves the cloud volume evaporates before reaching the ground (Fig. 15e and f).

Figure 16 shows vertical profiles of the anomalies of water vapor mixing ratio and vertical velocity in the cloud control volume (black box in Fig. 14) at 12 LT. Anomalies are defined as differences between the $x$- and $y$-averaged vertical profiles in the cloud volume (averaged on constant height levels) and the corresponding profiles in the surrounding environment. The stronger moistening in the s10 cases is clearly visible. This is in line with results by Griewank et al. (2022), who also diagnosed larger moisture anomalies in relatively wide convective updrafts in large-eddy simulations, but referring to the boundary layer over flat terrain (Southern Great Plains, US). They also documented larger vertical velocity anomalies in the wider updrafts, which contrasts with our findings in Fig. 16b. Griewank's anomaly profiles are derived from a population of 3D cloud objects, while we consider a specific cloud control volume spanning the whole domain length in $y$-direction. Thus, the observed discrepancies might result from the differences in the analysis methods, but possibly also from a fundamental difference between convection over flat and mountainous terrain.

According to Demko and Geerts (2010) the precipitation efficiency can be estimated as $P/VVF_b$ since $VVF_b$ can be seen as an estimate for the condensation (without evaporation) in the cloud.

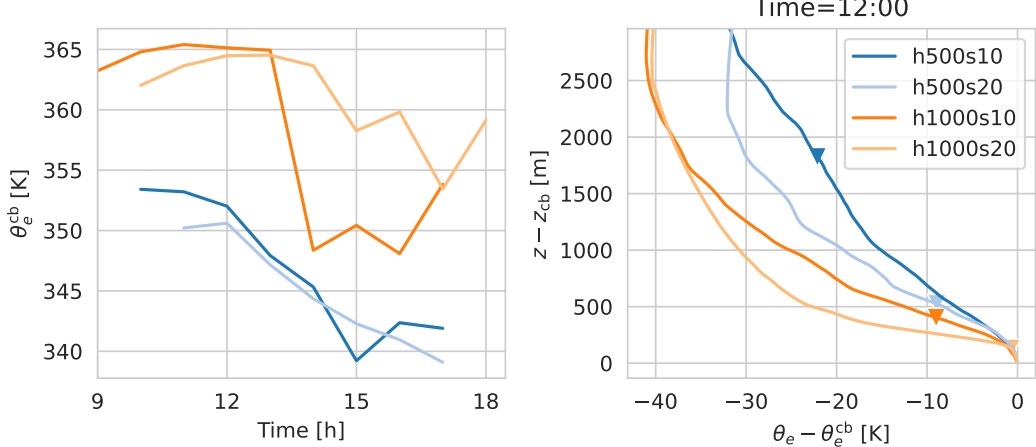

**Figure 17.** Equivalent potential temperature above the ridge ($x = 0$) conditionally averaged in $y$-direction for all grid points with vertically integrated cloud water content above 0.01 kg m$^{-2}$. Panel (a) shows time series at cloud base and (b) shows profiles at 12 LT relative to cloud base values. The triangles in (b) mark the top of the cloud control volume.

Using the accumulated values at the end of the simulation leads to a precipitation efficiency of 11.1 %, 0.03 %, 12.2 %, and 5.5 % for the h500s10, h500s20, h1000s10, and h1000s20 simulations, respectively. The h1000s10 can thus be called the most efficient in terms of precipitation production. The s20 simulations are very inefficient at producing precipitation. Demko and Geerts (2010) found hourly precipitation efficiencies of up to 1 % in the shallow convective period and up to 42 % for the deep convective period with an average of 9 %, which is roughly in line with our time-integrated values for the s10 mountains. However, the whole simulation setup and also the calculation of precipitation efficiency are rather different in their case, so a good agreement is not necessarily expected.

The precipitation efficiency can also be estimated from the water vapor microphysics tendency and precipitation alone, which requires fewer assumptions. We integrate the water vapor microphysics tendency in $y$-direction considering all gridpoints with positive net condensation, to exclude areas where evaporation dominates. Then, we further integrate the condensation rate in time, $x$, and $z$-direction. Dividing the total integrated (in space and time) precipitation by the integrated condensation yields the precipitation efficiency. Since the time integration considers 30-min intervals, possible compensating effects between condensation and evaporation on shorter time scales are neglected, leading to an overestimation of the precipitation efficiency. However, the relative magnitudes of the precipitation efficiency across different simulations should be approximately correct. With this method, we obtain a precipitation efficiency of 31.2 %, 0.06 %, 18.5 %, and 5.0 % for the h500s10, h500s20, h1000s10, and h1000s20 simulations, respectively. As expected, these numbers are higher than what we got from the cloud budget analysis. The h500s10 simulation is now by far the most efficient in producing precipitation. The s20 simulations have a much lower precipitation efficiency than the s10 runs, not only because of the lower total precipitation but also because of higher accumulated condensation.

To investigate the differences in moisture accumulation and turbulent entrainment between the s10 and the s20 runs further, we computed profiles of equivalent potential temperature $\theta_e$ above the ridge (at $x = 0$), conditionally averaged in the $y$-direction for all gridpoints with vertically integrated cloud water content above 0.01 kg m$^{-2}$. Figure 17b shows these profiles at 12 LT relative to the values at cloud base. We can clearly see that $\theta_e$ decreases more strongly with height in the s20 runs. Since $\theta_e$ is a conserved property in an ideal moist updraft unaffected by entrainment, this finding suggests that moist updrafts in the s20

simulations are subject to stronger entrainment of dry air. In addition, $\theta_e$ at cloud base (lower boundary of our cloud control volume) is slightly larger in the s10 runs, at least before the onset of DMC (Fig. 17a). Thus, moist updrafts in the s10 runs start out with a higher buoyancy potential, as already described in section 3.2.

       In summary, the cloud water budget provides additional evidence for why precipitation is delayed and reduced over the s20 mountains. First, less water vapor accumulates in the active cloud volume before precipitation begins, as demonstrated in

Sect. 3.2. Second, a smaller fraction of the water vapor that enters the cloud base is actually converted into precipitation, owing to increased entrainment of dry environmental air, as demonstrated here.

       Our findings are in line with those of other recent studies on convection initiation. Using idealized LES to study moist convection, Morrison et al. (2022) found that the bulk fractional entrainment rate, as well as the buoyancy dilution, increase with decreasing horizontal extent of the subcloud updraft. This implies that updrafts need to reach a minimum horizontal

extent for the transition from shallow to deep moist convection to occur. They also found that the environmental relative humidity strongly affects maximum cumulus height. Reduced entrainment is also the reason why supercell storms have greater longevity than other forms of organized convection. It was previously thought that resistance to entrainment in supercells is due to turbulence suppression in helical flow, but recent work demonstrated that their larger updraft width compared with ordinary convection is a more likely cause (Peters et al., 2020).

Our method for the cloud water budget analysis has some limitations. The budget is somewhat sensitive to the definition of the control volume, for instance, to the threshold value for the microphysics tendency. The time dependence of the control volume leads to an additional budget term for the volume change alone, which makes the whole budget hard to interpret, especially since we accumulate the budget components in time. In addition, the control volume spans the whole domain length in the $y$-direction. The analysis thus neglects the large heterogeneity in $y$ (Fig. 4), making it impossible to account for

entrainment in that direction. However, a fixed volume based on the maximum cloud extent would be even more problematic. For instance, the vapor flux at cloud base, VVF$_b$, would actually be sampled below cloud base for most of the time, because the cloud base is not stationary, just like the LCL (Fig. 6a). Additionally, the horizontal fluxes of water vapor and hydrometeors would be sampled at a great distance from the lateral cloud boundaries, because a fixed control volume would be much broader than the actual cloud region for most of the time (Fig. 13).

We are mainly interested in VVF$_b$, which determines the precipitation efficiency. This component of the cloud budget is evaluated accurately in our method because we adjust the cloud control volume to the actual cloud base, which is rather homogeneous in the $x$- and $y$-directions (Fig. 4 and Fig. 14).

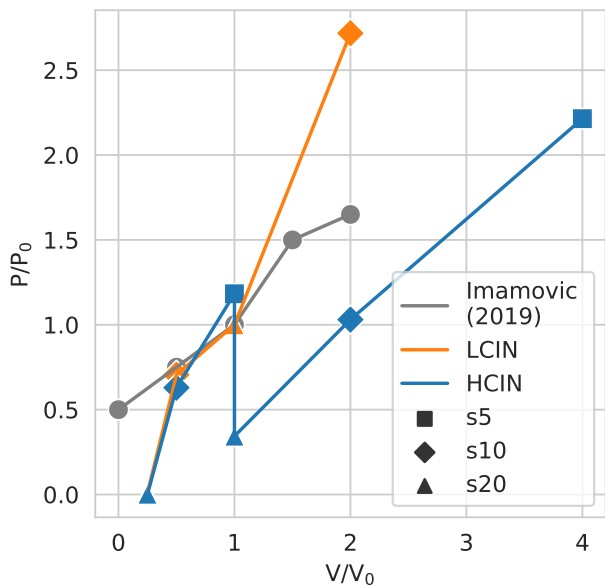

**Figure 18.** Total precipitation $P$ integrated in the $x$-direction as a function of terrain volume $V$ (equivalent to cross-sectional area) normalized by the values for the h1000s20 LCIN simulation ($P_0$ and $V_0$) for the runs with lower (LCIN) and higher (HCIN) initial CIN. The different markers denote the average slope (%) of the terrain. The gray line is based on Imamovic et al. (2019, their Fig. 7b).

### 4.3 Volume scaling

Figure 18 shows total precipitation $P$ integrated in the $x$-direction and normalized by $P_0$ as a function of normalized terrain volume $V/V_0$ for all LCIN and HCIN simulations. Following Imamovic et al. (2019), we choose a reference simulation for which $P = P_0$ and $V = V_0$ (h1000s20 LCIN in our case), and we evaluate how $P/P_0$ depends on $V/V_0$. Imamovic et al. (2019) found a linear relationship between $P/P_0$ and $V/V_0$ in a weakly inhibited environment (some of their results are reproduced in the gray line in Fig. 18). Our LCIN simulations show a monotonic relationship that is not far from linear. As Imamovic et al. (2019) speculated, this monotonic relationship does not hold in our strongly inhibited HCIN simulations. As discussed earlier, the s10 simulations ($V/V_0 = 0.5$ and 2) both produce more precipitation than the h1000s20 mountain ($V/V_0 = 1$) and the h500s5 mountain ($V/V_0 = 1$) produces more than the h1000s10 mountain ($V/V_0 = 2$). The effect of the slope angle seems to be more important than the effect of the mountain size in the HCIN simulations. Nevertheless, at constant slope angle, the larger mountains produce more precipitation than the smaller mountains, consistent with the theory of Imamovic et al. (2019) (see also Fig. 12).

Due to the low initial CIN in their simulations, Imamovic et al. (2019) still observed significant precipitation amounts for low mountain volumes and even for flat terrain, while we obtained hardly any precipitation for our smallest mountain ($V/V_0 = 0.25$, h500s20, both LCIN and HCIN).

# 5 Conclusions

We performed idealized LES simulations with the WRF model to study the effect of thermally-induced cross-valley circulations on convection initiation and strength for different mountain heights and widths, under synoptically undisturbed and convectively inhibited conditions. We found that mountain geometry has a significant impact on convection initiation time and strength. Steeper mountains lead to a later onset and lower intensity of deep moist convection despite the stronger associated thermally-driven circulation. The apparent discrepancy is explained by the behavior of convective updrafts on mountain tops, which are the focal points of the convective destabilization. Due to increased turbulent entrainment, the dry updrafts atop steeper mountains feature a smaller total moistening directly above the ridge. In addition, the updraft zone is narrower over the steeper mountains. This implies that total moistening around the mountain top is reduced further, while dilution of the saturated updrafts by entrainment of dry, non-cloudy air is enhanced.

We obtained these results considering mountains with heights of 500 and 1000 m, average slopes of 10 and 20 %, and a nearly isothermal temperature profile at all heights up to 500 m above the mountaintop (and conditional instability further aloft). However, we verified that the findings are robust also for 5 % average slope, for a more unstable stratification, and also for slightly higher and steeper mountains.

Classical convective indices like LFC, CIN, and CAPE show a faster destabilization over the less steep mountains due to increased moisture content. The analysis of the circulation and the water vapor and total cloud water budgets above the ridge provide valuable insights into the dynamical processes that determine moisture availability near mountain tops.

The moisture tendency at the mountaintop is not driven purely by the strength of the mean circulation. Actually, before condensation starts, moistening due to the convergence of advective fluxes is almost entirely compensated by drying due to turbulent entrainment. The updrafts in the column directly above the ridge of the steeper mountains gain more moisture by mean moisture convergence. However, due to increased turbulent entrainment, the total moistening is lower over the steeper mountains. In addition, their updraft zone is also narrower, leading to even stronger differences in moisture accumulation when the whole updraft zone is considered. When clouds start to form, the circulation generates a higher net advective tendency (mean + turbulent), which is then compensated partly by condensation.

Based on Demko and Geerts (2010), we developed a novel approach to quantify the components of the water budget for a convective cloud using the output of the online tendency and flux averaging tool WRFlux (Göbel et al., 2022). The extent of the cloud is defined using a threshold for the net condensation rate to exclude inactive parts of the cloud. In our simulations, the saturated updrafts over less steep mountains gain more moisture from the vapor flux at cloud base leading to significantly higher moisture accumulation and their equivalent potential temperature decreases less strongly with height above cloud base, suggesting a lower entrainment rate compared with the steeper simulations. The precipitation efficiency, a measure for how much of the condensed water eventually precipitates, as derived from the cloud water budget and alternatively from the water vapor microphysics tendency, is also considerably larger over the less steep mountains. This is not only due to the higher total precipitation but also due to lower total condensation than over the steeper mountains.

In the simulations with lower initial convective inhibition we could confirm the nearly linear scaling between accumulated precipitation amount and mountain volume documented by Imamovic et al. (2019). In contrast, under highly inhibited conditions, the precipitation amount is apparently controlled more by mountain steepness than by mountain size. In these strongly inhibited regimes, we could not confirm the linear or even a monotonic scaling. However, if the slope angle is held constant, a monotonic relationship between precipitation and mountain volume (controlled by mountain height alone) does occur.

In this study, we only investigated highly idealized orographies. As a next step, semi-idealized simulations with realistic orography as in Weinkaemmerer et al. (2023) and heterogeneous land cover or soil moisture can provide further valuable insights into the sensitivity of convection to mountain geometry. The interactions between cross-valley circulations, valley winds, plain-to-mountain winds, and an upper-level, large-scale wind add to the complexity that needs to be investigated with 3D orographies. The background wind impact might be the most critical. Intuitively, large-scale advection can either enhance or counteract moisture accumulation at mountain tops, depending on how moist or dry the incoming air is. In any case, it would tend to make the horizontal moisture distribution less dependent on the underlying orography, possibly canceling the steepness effect we documented.

To connect insights from idealized simulations with real-world processes, an analysis of the terrain geometry around hotspots in convection initiation climatologies (e.g., from radar reflectivity or lightning frequency) can also be investigated.

Our results have a possibly important implication. If vertical mass transport at mountain tops is, all other factors equal, systematically affected by orographic sharpness, then it is nearly impossible for operational-scale numerical weather prediction models to resolve it correctly. In fact, the model orography is invariably much more regular than the actual one, not only because it is sampled at discrete points, but also because it is artificially smoothed to prevent numerical instability. Thus, novel approaches for the parameterization of boundary-layer transport over mountains, accounting for sub-grid-scale orographic complexity, are likely needed.

## Appendix A: Vertical grid

The vertical grid in our $\Delta x = 50$ m simulations consists of two layers comprising in total $n_z$ grid points. In the first layer, $0 < z < z_1$, vertical grid increments $\Delta z_i$ increase smoothly from $\Delta z_{\min} = 20$ m to $\Delta z_{\max} = 100$ m, as prescribed by:

$$\Delta z_{\mathrm{i}} = \Delta z_{\mathrm{m}} + \frac{\Delta z_{\min} - \Delta z_{\mathrm{m}}}{\tanh(2)} \tanh\left(4\frac{2\mathrm{i} - n_z}{2 - n_z}\right) \tag{A1}$$

for $i = 1, ..., n_z - 1$ with

$$\Delta z_{\mathrm{m}} = \frac{\Delta z_{\min} + \Delta z_{\max}}{2}. \tag{A2}$$

In the second layer, $z_1 < z < z_{\mathrm{top}}$, $\Delta z_{\mathrm{i}} = \Delta z_{\max}$. This definition is inspired by one of the options available in the ARPS model (Xue et al., 2000) and leads to a strong increase of $\Delta z$ with height and leveling off towards $z_1$.

Since the $\eta$ vertical coordinate in WRF is pressure-based, we transform the nominal height levels $z_i$ to nominal pressure levels $p_i$ assuming hydrostatic balance in the initial potential temperature profile.

Further, we convert to nominal $\eta_i$ terrain-following levels with $\eta_i = (p_i - p_{\text{top}})/(p_s - p_{\text{top}})$, using $p_s = p_0 = 1000$ hPa for the surface pressure. These $\eta_i$ values span the $[0,1]$ range. Because $p_s$ varies markedly with terrain elevation and is generally smaller than $p_0$, a given $\Delta\eta_i$ corresponds to a smaller height increment over elevated terrain. The $\eta$ coordinate is hybrid terrain-following, becoming isobaric above height $z_{\text{hyb}}$.

In the moist simulations $n_z = 253$, $z_1 = 12$ km, $z_{\text{top}} = 17$ km, and $z_{\text{hyb}} = 10$ km. In the dry simulations $n_z = 102$, $z_1 = 3$ km, $z_{\text{top}} = 8$ km and $z_{\text{hyb}} = 6$ km.

*Code and data availability.* The $y$-averaged model output, the unaveraged output at the ridge, the namelist and input sounding files, and the code to reproduce the simulations, the figures, and the input soundings are available on Zenodo (Göbel, 2023a). The simulations can be reproduced using WRFlux v1.6.0 (Göbel, 2023b) and the modified source code files from Göbel (2023a).

*Author contributions.* MG developed the concept, ran and analyzed the simulations, and prepared the manuscript. StS and MWR supervised the work and reviewed and edited the paper extensively. StS acquired the funding and administers the project.

*Competing interests.* The authors declare that they have no conflict of interest.

*Acknowledgements.* This work is funded by the Austrian Science Fund (FWF) research project P30808-N32 "Multiscale Interactions in Convection Initiation in the Alps". The computational results presented have been achieved using the Vienna Scientific Cluster (VSC) and the LEO cluster of the University of Innsbruck. Matthias Göbel thanks Wiebke Scholz for her support and the various fruitful discussions. Last but not least, we thank the two anonymous reviewers for their valuable comments and suggestions.

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
