# Peer review of "Adverse impact of terrain steepness on thermally-driven initiation of orographic convection"

_EGUsphere, 2023_

## Author Response (AR1)

**Answer to RC1**

**Major comments**

**1.**

I found the cloud-water budget analysis of section 4.2 to be rather complicated and not as insightful as it could have been. I particularly wonder why the authors chose to make the lateral boundaries of the control volume vary in time. Normally this is avoided, and it adds a term to the budget with a comparable magnitude to the other large terms. Why not just fix the boundaries of the control volume based on the mountain dimensions, as done by Demko and Geerts (2010)? That would be much simpler, and it would help to expose the processes that matter the most without interference from the changing control volume.

**We tried out and compared several ways of defining the cloud volume including a stationary definition. Finally, we decided to use a time-dependent control volume because the cloud size varies a lot in time.**

**We acknowledge that the time dependence of the cloud volume makes the whole analysis rather complicated and may cause problems in the interpretation, especially because we accumulate the budget terms over time. However, we think that a fixed control volume would cause even bigger problems in the interpretation.**

**If we chose a rather large control volume and kept it constant, then in the early stages of the cloud development (when the cloud is still very narrow) the subcloud moisture flux (VVF_b) would include subsidence and updraft regions away from the cloud itself and thus be harder to interpret. Also, the cloud base rises with time in our simulations. The lower boundary of the control volume could be set to the lowest occurring cloud base in the simulation (in the early stage), but this choice would make the boundary lie within the region where precipitation evaporates below cloud base later in the day. This clearly introduces problems in the interpretation of the budget components.**

**Demko and Geerts (2010) fixed only the horizontal boundaries of the cloud control volume. Their vertical boundaries follow the evolution of the PBL height. Thus, their control volume also changes with time.**

**We added a discussion of the limitations of the cloud budget analysis (L455-467) and, for comparison, also included a simpler,**

**alternative method for computing precipitation efficiency from precipitation and water vapor microphysics tendency alone (L423-434).**

**2.**

Beyond the above concern, there is limited insight into cloud-layer processes. The budget has value, but ultimately it is only used to show that (i) the less-steep mountains cause larger moisture fluxes into the cloud layer due to their wider updrafts, which follows directly from section 3.2 (which, by contrast, was more rigorous), and (ii) the clouds that form over the steeper mountains tend to evaporate rather than making much precipitation. From this result, the authors conclude that the narrowness of the clouds over the steeper mountains is what prevents them from intensifying. The clouds do look narrower in Fig. 3 but that result isn't confirmed quantitatively. Also, the role of cloud width on cloud life cycles is not necessarily settled scientifically. It is a hypothesis that has been refuted in some studies (e.g., Dawe and Austin, ACP 2013). Therefore, I am not convinced that the authors' conclusion is justified. Fortunately, there are many useful analyses that one can do to gain insight into cloud processes. I'll start with the simplest: conditionally average (in the horizontal) a conserved variable (e.g., moist static energy or equivalent potential temperature) within the clouds and compare the profiles between different cases. This would show whether the clouds over steeper mountains (i) originate with lower moist static energy (or theta_e), and how they evolve with height. My guess is that these variables are already smaller at cloud base, and decrease faster with height, over steeper mountains, which would imply reduced buoyancy potential and faster dilution with height.

**We are thankful for this very useful comment.**

**In Fig. 13 (was Fig. 12 before) it is evident that the cloud (or rather the cloud zone, since there can be several clouds next to each other) is indeed narrower in the s20 runs.**

**As suggested, we computed equivalent potential temperature profiles above the ridge conditionally averaged in y for cloudy columns and added this as Fig. 17. Before precipitation onset, theta_e is indeed smaller at cloud base and decreases faster with height for the s20 simulations compared with the respective s10 simulations (discussion of plot at L435-442).**

**3.**

The orographic clouds over the steeper mountains are so narrow in Fig. 3 that they may be under-resolved. Looking closely at the figure, the cloud appears to be < 500 m wide. With a horizontal grid spacing of 100 m, this cloud would be poorly resolved even in an inviscid model. But WRF is highly diffusive, with an effective resolution of 5-10dx, which means the cloud is poorly resolved. Thus, the cloud is likely subject to strong enough numerical diffusion to affect its development. If the authors have the resources, I suggest redoing the steeper mountain cases at 50 m or run a third case with an even wider mountain at 100 m. The latter would allow you to compare two cases (s10 and, say, s5) where the slope differs but the clouds are well resolved in both cases. This would also be useful for showing the robustness of the experimental trends, beyond the two cases (for a given peak height) currently conducted.

**We reran all simulations at 50 m grid spacing and adapted the manuscript accordingly. The results are very similar. In these simulations, the stronger moisture accumulation above the ridge in the s10 runs becomes even more clear. We also included two s5 simulations in the discussion section (Fig. 12)**

**Minor comments**

**1.**

L. 76: I'm not sure if the "local thermodynamic profile" is the most appropriate here. I would suggest replacing it with the "environmental thermodynamic profile", because the orography locally changes the profile. If we're thinking of a simple cause and effect relation, it is better to separate the environmental flow from the locally modified orographic flow.

**OK, done**

**2.**

L. 117-118: I find the motivation of this study to be rather vague and not very exciting: "we continue investigating the impact of terrain geometry on orographic CI and take a closer look at strongly inhibited conditions". This objective says what the authors intend to do, not the scientific problem they intend to address. It is better to frame an objective around a scientific question, because the methodology becomes a step toward achieving that goal. The methodology should not be the end unto itself.

**In the introduction, we discuss the various controlling factors for CI in mountainous terrain. Most of these factors have been**

**investigated already. However, the terrain geometry has not been studied as much, especially not under strongly inhibited conditions.**

**We added a sentence to make the motivation clearer (L121-123).**

**3.**

L. 135-136: I find it a bit odd that you would use positive-definite advection for scalars and WENO for vectors. Normally one would use WENO on scalar advection to avoid spurious oscillations that degrade the simulation of clouds. Can you please justify these choices? I'm guessing you used positive definite advection to ensure moisture conservation, which makes sense, but the monotonic and WENO options also do quite well at conserving moisture (I believe). If you see spurious oscillations forming near cloud edges in your simulations, it is likely owing to the PD scheme used for scalars.

**Previous tests with WRFlux showed that the (dry) potential temperature budget cannot be adequately closed when using the WENO or monotonic advection option for moisture together with the moist potential temperature formulation due to conservation issues. However, since we are only interested in the moisture budget in this study, we switched to the WENO option for scalars in the new dx=50m simulations.**

**4.**

L. 231-232: the authors discuss the relative humidity diurnal evolution, but they don't seem to show it. When discussing quantities are not shown, the reader should be informed of this via "(not shown)" or similar.

**We deleted this part. As suggested by another reviewer, we computed convective indices for mixed-layer (instead of surface) parcels and modified the text to reflect the new findings.**

**5.**

Figures 7-8: The authors show these nice budgets but don't clarify where the budgets are measured, which is a pity. The authors only say that they are "above the ridge", which is rather vague to me. Do you mean the grid point over the mountain crest? Please specify that, because it's critical context.

**The budgets are measured in the column directly above the ridge (x=0). We added this info in the figure caption and in the text.**

**As written in the text (L264), in Eq. 2, and in the caption of Fig. 7 (now Fig. 8) the budgets are vertically integrated, so all grid points at x=0 between the surface and the model top are considered.**

**6.**

L. 267: "final" -> total

**OK, done**

**7.**

L. 281-288: Are you confident that Fig. 9 allows you to calculate updraft width? It concerns me that the authors are measuring mass flux over the entire column and using that evaluate the width of the surface-based updraft. What if there is compensating mass flux aloft and/or cumulus convection? Both of these are not directly related to the surface updraft width but would figure into the column-integrated mass flux. On L. 288, the authors state that cloud latent-heat release leads to a widening of the updrafts. I don't think that conclusion is justified, because the authors make no attempt to distinguish dry and moist updrafts in this calculation. It could be that the dry updrafts don't change but moist convection aloft changes the mass flux profiles. This is an example of the authors reaching conclusions without acknowledging the limits of their analyses (as in major comment #2).

**We changed the computation for Fig. 9b (now Fig. 10b). The integral of the vertical mass flux is now only over the dry part up to cloud base. The width of the moist part is shown in the cloud budget analysis (now Fig. 13) instead.**

**The conclusion remains the same: the updraft zone is wider for the s10 runs and the onset of cloud development leads to a significant widening of this updraft zone.**

**We added a note, that the individual updrafts in y-direction are not necessarily wider in the s10 runs, but that they are more frequent and/or stronger up to some distance from the ridge compared with the s20 runs, which leads to a wider y-averaged updraft zone in Fig. 10 (L290-293).**

**8.**

L. 358: To me at least, the term Epbl in the moisture budget seemed irrelevant at first, since it relates to evaporation outside the control

volume. I was only able to understand its value by realizing that the authors are probably only using it to infer, based only on measurements of *surface* precipitation (P) and microphysical tendencies, how much precipitation sediments out of the control volume, which is P+Epbl. The authors do not discuss this point, which I think invites confusion. As I mentioned in major comment 1, I think the complexity of the deformable control volume is a shortcoming, and this confusion compounds that effect.

**We changed the definition of the precipitation term P to be the precipitation that falls out of the control volume, so surface precipitation+E_PBL, to limit confusion. E_PBL is thus not part of the cloud budget anymore and is only plotted for information. We added some sentences to make the role of E_PBL clearer (L374-375 and L402-403).**

**9.**

L. 378-379: Related to major comment 2, the authors seem to draw this conclusion out of thin air. They claim that the moist updrafts over the less-steep mountains are wider than those over steeper mountains, which hasn't been quantitatively shown. They then conclude that these updrafts are more susceptible to entrainment, which they also haven't shown…but I agree is indirectly implied by the moisture budget. Could it just be that these updrafts, due to lower moisture content at cloud base, are unable to generate as much adiabatic buoyancy, and are thus more susceptible to any suppressive effect (entrainment, detrainment, adverse pressure perturbations, etc.)? Decreased initial updraft moisture over steeper isladns was suggested by the dry moisture budget of the subcloud layer. This just seems like a conclusion that is stated much more strongly than is justified by the data, and alternative hypotheses are not considered.

**The width of the dry updrafts is shown in Fig. 10, while the width of the cloudy part of the updraft is shown in Fig. 13. Both show a wider updraft zone for the s10 runs.**

**As written above, we added Fig. 17 (theta_e profiles) to show that the s10 runs have a higher theta_e at cloud base and that this theta_e also decreases less strongly with height, hinting at weaker dilution compared with the s20 runs.**

**10.**

L. 422: The authors must appreciate that moist and dry updrafts behave differently and probably should not be lumped together. This sentence refers to narrower updrafts losing "moisture and cloud droplets", not

specifying whether the updrafts are dry or saturated. The presence of cloud droplets assumes saturated updrafts, but "thermal plumes" earlier in the sentence implies dry updrafts. Which one is it, or is this statement general enough for both? This problem reappears on L. 438, where the term "convective updrafts" is used without specifying dry or saturated.

**We reformulated those sentences.**

**11.**

L. 443-444: I agree with this conclusion, but I think you could elaborate on it a bit. The fact that you have shown that you can reproduce Imamovic et al's (2019) volume scaling at lower CIN gives you much more credibility in declaring the conditions under which it doesn't work. I think that point is worth highlighting because it strengthens your conclusion.

**OK, done (L516-520).**

**We also changed the figure (now Fig. 18). Precipitation is now integrated instead of averaged in the x-direction. The x-average made the comparison between simulations of different domain widths problematic. We also included the new s5 simulations.**

**12.**

L. 449: A pet peeve of mine is the description of orography as "3D". How can orography be 3D when it only varies at most in two directions (x and y)? A function of x and y is 2D, not 3D.

**In the mathematical sense of a 2D function, you're right. We believe, however, that the term 2D would be even more confusing since that type of orography corresponds to the 3D world that we live in. Thus, we just deleted the word.**

**Answer to RC2**

**Moderate concerns:**

**1.**
The primary conclusion is that wider terrain leads to stronger convection. This is tied to the initial updrafts being wider and less resistant to entrainment. It was not immediately clear in reading the manuscript why the wider terrain leads to wider updrafts. Perhaps this is common knowledge in mountain meteorology and I am not aware of the reason,

but it would be nice to make the connection obvious in the paper. If there isn't a well-established reason for this, it could be helpful to speculate as to why.

**Our explanations concerning the wider updrafts were probably misleading. We were referring to the wider updraft zone, visible in the y-averaged output (Fig.10). This does not necessarily mean that the individual updrafts are wider, but that they occur more frequently or are stronger in the vicinity of the mountaintop compared with other simulations.**
**In all our simulations, static stability is higher at low levels, over the valley. Therefore, updraft development is favored at higher altitudes, over the ridges. Intuitively, the s10 terrain provides a larger area above a certain height than the s20 terrain, offering more room for thermals to develop, which ultimately leads to wider updraft zones.**
**We tried to make this concept clearer in the revised manuscript (L290-298).**
**However, we are not aware of any publications that studied this issue in depth.**

**2.**
The horizontal grid spacing for the main simulation suite that the manuscript is based on is largely in line with that of other recent manuscripts (e.g., Morrison et al. 2021, Peters et al. 2020). While this might be on the marginal side for resolving the influence of entrainment on smaller clouds/thermals like the ones depicted here, I have little reason to believe that the results of the study would change if a finer resolution were to be used. The runs at 250 m grid spacing contain clouds that are marginally resolved; however, the conclusions drawn from these simulations also fit within the conceptual framework that the manuscript develops, and I also think that the conclusions drawn from these sensitivity tests would not change at an increased spatial resolution.

**As written in the reply to major comment 3 of reviewer 1, we reran all dx = 100 simulations with dx = 50 m to avoid under-resolved updrafts, especially over the steeper mountains.**

**3.**
The authors may wish to discuss some of their results in the context of recent publications investigating updraft width and entrainment and their eventual influence on deep convection initiation including Morrison et al. (2021) and Peters et al. (2020).

**Thank you for the references, they are indeed very relevant here. We included them in the discussion section (L447-454)**

**Minor comments**

**1.**
Line 7: I found the use of relatively steep and moderately steep confusing throughout the manuscript and had to keep reminding myself of which was which. Perhaps more clear language could be used like steeper and less steep? I am open to leaving it as is, or leaning more on the s10 and s20 references, but did want to note that it was confusing to me.
**OK, we use s10 and s20 at most places now. Only in the abstract and conclusion do we use "steep" and "less steep".**

**2.**
Line 225: Would calculating these convective indices with mixed layer parcel properties (generally calculated over the lowest 100 mb) be more predictive? Sometimes these are used to account for entrainment, although they aren't always perfect. I am not very familiar with this parameter, but given the focus on entrainment, using entraining convective available potential energy (ECAPE, https://arxiv.org/abs/2301.04712) may be predictive here. I'm not sure how useful it is though, given there us very little storm relative flow here. If you do not think this would be applicable or useful, I am fine ignoring this suggestion.

**Thank you for this comment. The mixed-layer parcel indeed seems much more useful/predictive here. We changed the corresponding plot and the text (L9-11, L231-249, and L497-498). Thus, we don't see the need for implementing the complicated ECAPE definition, also because of the missing background flow in our simulations.**